# COLA: Decentralized Linear Learning

**Lie He**[*]
EPFL
lie.he@epfl.ch

**An Bian**[*]
ETH Zurich
ybian@inf.ethz.ch

**Martin Jaggi**
EPFL
martin.jaggi@epfl.ch

## Abstract

Decentralized machine learning is a promising emerging paradigm in view of global challenges of data ownership and privacy. We consider learning of linear classification and regression models, in the setting where the training data is decentralized over many user devices, and the learning algorithm must run on-device, on an arbitrary communication network, without a central coordinator. We propose COLA, a new decentralized training algorithm with strong theoretical guarantees and superior practical performance. Our framework overcomes many limitations of existing methods, and achieves communication efficiency, scalability, elasticity as well as resilience to changes in data and allows for unreliable and heterogeneous participating devices.

## 1 Introduction

With the immense growth of data, decentralized machine learning has become not only attractive but a necessity. Personal data from, for example, smart phones, wearables and many other mobile devices is sensitive and exposed to a great risk of data breaches and abuse when collected by a centralized authority or enterprise. Nevertheless, many users have gotten accustomed to giving up control over their data in return for useful machine learning predictions (e.g. recommendations), which benefits from joint training on the data of all users combined in a centralized fashion.

In contrast, decentralized learning aims at learning this same global machine learning model, without any central server. Instead, we only rely on distributed computations of the devices themselves, with each user's data never leaving its device of origin. While increasing research progress has been made towards this goal, major challenges in terms of the privacy aspects as well as algorithmic efficiency, robustness and scalability remain to be addressed. Motivated by aforementioned challenges, we make progress in this work addressing the important problem of training generalized linear models in a fully decentralized environment.

Existing research on decentralized optimization, $\min_{\mathbf{x} \in \mathbb{R}^n} F(\mathbf{x})$, can be categorized into two main directions. The seminal line of work started by Bertsekas and Tsitsiklis in the 1980s, cf. [Tsitsiklis et al., 1986], tackles this problem by splitting the parameter vector $\mathbf{x}$ by coordinates/components among the devices. A second more recent line of work including e.g. [Nedic and Ozdaglar, 2009, Duchi et al., 2012, Shi et al., 2015, Mokhtari and Ribeiro, 2016, Nedic et al., 2017] addresses sum-structured $F(\mathbf{x}) = \sum_k F_k(\mathbf{x})$ where $F_k$ is the local cost function of node $k$. This structure is closely related to empirical risk minimization in a learning setting. See e.g. [Cevher et al., 2014] for an overview of both directions. While the first line of work typically only provides convergence guarantees for smooth objectives $F$, the second approach often suffers from a "lack of consensus", that is, the minimizers of $\{F_k\}_k$ are typically different since the data is not distributed i.i.d. between devices in general.

---

[*]These two authors contributed equally

**Contributions.** In this paper, our main contribution is to propose COLA, a new decentralized framework for training generalized linear models with convergence guarantees. Our scheme resolves both described issues in existing approaches, using techniques from primal-dual optimization, and can be seen as a generalization of COCOA [Smith et al., 2018] to the decentralized setting. More specifically, the proposed algorithm offers

- *Convergence Guarantees:* Linear and sublinear convergence rates are guaranteed for strongly convex and general convex objectives respectively. Our results are free of the restrictive assumptions made by stochastic methods [Zhang et al., 2015, Wang et al., 2017], which requires i.i.d. data distribution over all devices.

- *Communication Efficiency and Usability:* Employing a data-local subproblem between each communication round, COLA not only achieves communication efficiency but also allows the re-use of existing efficient single-machine solvers for on-device learning. We provide practical decentralized primal-dual certificates to diagnose the learning progress.

- *Elasticity and Fault Tolerance:* Unlike sum-structured approaches such as SGD, COLA is provably resilient to changes in the data, in the network topology, and participating devices disappearing, straggling or re-appearing in the network.

Our implementation is publicly available under github.com/epfml/cola .

## 1.1 Problem statement

**Setup.** Many machine learning and signal processing models are formulated as a composite convex optimization problem of the form

$$\min_{\mathbf{u}} \; l(\mathbf{u}) + r(\mathbf{u}),$$

where $l$ is a convex loss function of a linear predictor over data and $r$ is a convex regularizer. Some cornerstone applications include e.g. logistic regression, SVMs, Lasso, generalized linear models, each combined with or without L1, L2 or elastic-net regularization. Following the setup of [Dünner et al., 2016, Smith et al., 2018], these training problems can be mapped to either of the two following formulations, which are dual to each other

$$\min_{\mathbf{x} \in \mathbb{R}^n} \left[ F_A(\mathbf{x}) := f(\mathbf{A}\mathbf{x}) + \sum_i g_i(x_i) \right] \tag{A}$$

$$\min_{\mathbf{w} \in \mathbb{R}^d} \left[ F_B(\mathbf{w}) := f^*(\mathbf{w}) + \sum_i g_i^*(-\mathbf{A}_i^\top \mathbf{w}) \right], \tag{B}$$

where $f^*, g_i^*$ are the convex conjugates of $f$ and $g_i$, respectively. Here $\mathbf{x} \in \mathbb{R}^n$ is a parameter vector and $\mathbf{A} := [\mathbf{A}_1; \ldots; \mathbf{A}_n] \in \mathbb{R}^{d \times n}$ is a data matrix with column vectors $\mathbf{A}_i \in \mathbb{R}^d, i \in [n]$. We assume that $f$ is smooth (Lipschitz gradient) and $g(\mathbf{x}) := \sum_{i=1}^n g_i(x_i)$ is *separable*.

**Data partitioning.** As in [Jaggi et al., 2014, Dünner et al., 2016, Smith et al., 2018], we assume the dataset $\mathbf{A}$ is distributed over $K$ machines according to a partition $\{\mathcal{P}_k\}_{k=1}^K$ of the *columns* of $\mathbf{A}$. Note that this convention maintains the flexibility of partitioning the training dataset either by samples (through mapping applications to (B), e.g. for SVMs) or by features (through mapping applications to (A), e.g. for Lasso or L1-regularized logistic regression). For $\mathbf{x} \in \mathbb{R}^n$, we write $\mathbf{x}_{[k]} \in \mathbb{R}^n$ for the $n$-vector with elements $(\mathbf{x}_{[k]})_i := x_i$ if $i \in \mathcal{P}_k$ and $(\mathbf{x}_{[k]})_i := 0$ otherwise, and analogously $\mathbf{A}_{[k]} \in \mathbb{R}^{d \times n_k}$ for the corresponding set of local data columns on node $k$, which is of size $n_k = |\mathcal{P}_k|$.

**Network topology.** We consider the task of joint training of a global machine learning model in a decentralized network of $K$ nodes. Its connectivity is modelled by a mixing matrix $\mathcal{W} \in \mathbb{R}_+^{K \times K}$. More precisely, $\mathcal{W}_{ij} \in [0, 1]$ denotes the connection strength between nodes $i$ and $j$, with a non-zero weight indicating the existence of a pairwise communication link. We assume $\mathcal{W}$ to be symmetric and doubly stochastic, which means each row and column of $\mathcal{W}$ sums to one.

The spectral properties of $\mathcal{W}$ used in this paper are that the eigenvalues of $\mathcal{W}$ are real, and $1 = \lambda_1(\mathcal{W}) \geq \cdots \geq \lambda_n(\mathcal{W}) \geq -1$. Let the second largest magnitude of the eigenvalues of $\mathcal{W}$ be $\beta := \max\{|\lambda_2(\mathcal{W})|, |\lambda_n(\mathcal{W})|\}$. $1 - \beta$ is called the *spectral gap*, a quantity well-studied in graph theory and network analysis. The spectral gap measures the level of connectivity among nodes. In the extreme case when $\mathcal{W}$ is diagonal, and thus an identity matrix, the spectral gap is 0 and there is no communication among nodes. To ensure convergence of decentralized algorithms, we impose

the standard assumption of positive spectral gap of the network which includes all connected graphs, such as e.g. a ring or 2-D grid topology, see also Appendix B for details.

## 1.2    Related work

Research in decentralized optimization dates back to the 1980s with the seminal work of Bertsekas and Tsitsiklis, cf. [Tsitsiklis et al., 1986]. Their framework focuses on the minimization of a (smooth) function by distributing the components of the parameter vector $\mathbf{x}$ among agents. In contrast, a second more recent line of work [Nedic and Ozdaglar, 2009, Duchi et al., 2012, Shi et al., 2015, Mokhtari and Ribeiro, 2016, Nedic et al., 2017, Scaman et al., 2017, 2018] considers minimization of a sum of individual local cost-functions $F(\mathbf{x}) = \sum_i F_i(\mathbf{x})$, which are potentially non-smooth. Our work here can be seen as bridging the two scenarios to the primal-dual setting (A) and (B).

While decentralized optimization is a relatively mature area in the operations research and automatic control communities, it has recently received a surge of attention for machine learning applications, see e.g. [Cevher et al., 2014]. Decentralized gradient descent (DGD) with diminishing stepsizes was proposed by [Nedic and Ozdaglar, 2009, Jakovetic et al., 2012], showing convergence to the optimal solution at a sublinear rate. [Yuan et al., 2016] further prove that DGD will converge to the neighborhood of a global optimum at a linear rate when used with a constant stepsize for strongly convex objectives. [Shi et al., 2015] present EXTRA, which offers a significant performance boost compared to DGD by using a gradient tracking technique. [Nedic et al., 2017] propose the DIGing algorithm to handle a time-varying network topology. For a static and symmetric $\mathcal{W}$, DIGing recovers EXTRA by redefining the two mixing matrices in EXTRA. The dual averaging method [Duchi et al., 2012] converges at a sublinear rate with a dynamic stepsize. Under a strong convexity assumption, decomposition techniques such as decentralized ADMM (DADMM, also known as consensus ADMM) have linear convergence for time-invariant undirected graphs, if subproblems are solved exactly [Shi et al., 2014, Wei and Ozdaglar, 2013]. DADMM+ [Bianchi et al., 2016] is a different primal-dual approach with more efficient closed-form updates in each step (as compared to ADMM), and is proven to converge but without a rate. Compared to CoLa, neither of DADMM and DADMM+ can be flexibly adapted to the communication-computation tradeoff due to their fixed update definition, and both require additional hyperparameters to tune in each use-case (including the $\rho$ from ADMM). Notably CoLa shows superior performance compared to DIGing and decentralized ADMM in our experiments. [Scaman et al., 2017, 2018] present lower complexity bounds and optimal algorithms for objectives in the form $F(\mathbf{x}) = \sum_i F_i(\mathbf{x})$. Specifically, [Scaman et al., 2017] assumes each $F_i(\mathbf{x})$ is smooth and strongly convex, and [Scaman et al., 2018] assumes each $F_i(\mathbf{x})$ is Lipschitz continuous and convex. Additionally [Scaman et al., 2018] needs a boundedness constraint for the input problem. In contrast, CoLa can handle non-smooth and non-strongly convex objectives (A) and (B), suited to the mentioned applications in machine learning and signal processing. For smooth nonconvex models, [Lian et al., 2017] demonstrate that a variant of decentralized parallel SGD can outperform the centralized variant when the network latency is high. They further extend it to the asynchronous setting [Lian et al., 2018] and to deal with large data variance among nodes [Tang et al., 2018a] or with unreliable network links [Tang et al., 2018b]. For the decentralized, asynchronous consensus optimization, [Wu et al., 2018] extends the existing PG-EXTRA and proves convergence of the algorithm. [Sirb and Ye, 2018] proves a $O(K/\epsilon^2)$ rate for stale and stochastic gradients. [Lian et al., 2018] achieves $O(1/\epsilon)$ rate and has linear speedup with respect to number of workers.

In the distributed setting with a central server, algorithms of the CoCoA family [Yang, 2013, Jaggi et al., 2014, Ma et al., 2015, Dünner et al., 2018]—see [Smith et al., 2018] for a recent overview— are targeted for problems of the forms (A) and (B). For convex models, CoCoA has shown to significantly outperform competing methods including e.g., ADMM, distributed SGD etc. Other centralized algorithm representatives are parallel SGD variants such as [Agarwal and Duchi, 2011, Zinkevich et al., 2010] and more recent distributed second-order methods [Zhang and Lin, 2015, Reddi et al., 2016, Gargiani, 2017, Lee and Chang, 2017, Dünner et al., 2018, Lee et al., 2018].

In this paper we extend CoCoA to the challenging *decentralized* environment—with no central coordinator—while maintaining all of its nice properties. We are not aware of any existing primal-dual methods in the decentralized setting, except the recent work of [Smith et al., 2017] on federated learning for the special case of multi-task learning problems. Federated learning was first described by [Konecný et al., 2015, 2016, McMahan et al., 2017] as decentralized learning for on-device learning applications, combining a global shared model with local personalized models. Current

---

**Algorithm 1:** COLA: **Co**mmunication-Efficient Decentralized **L**inear Le**a**rning

---
1    **Input**: Data matrix $\mathbf{A}$ distributed column-wise according to partition $\{\mathcal{P}_k\}_{k=1}^K$. Mixing matrix $\mathcal{W}$. Aggregation parameter $\gamma \in [0, 1]$, and local subproblem parameter $\sigma'$ as in (1). Starting point $\mathbf{x}^{(0)} := \mathbf{0} \in \mathbb{R}^n, \mathbf{v}^{(0)} := \mathbf{0} \in \mathbb{R}^d, \mathbf{v}_k^{(0)} := \mathbf{0} \in \mathbb{R}^d \; \forall \; k = 1, \ldots K$;

2    **for** $t = 0, 1, 2, \ldots, T$ **do**
3      **for** $k \in \{1, 2, \ldots, K\}$ **in parallel over all nodes do**
4        compute locally averaged shared vector $\mathbf{v}_k^{(t+\frac{1}{2})} := \sum_{l=1}^K \mathcal{W}_{kl} \mathbf{v}_l^{(t)}$
5        $\Delta \mathbf{x}_{[k]} \leftarrow \Theta$-approximate solution to subproblem (1) at $\mathbf{v}_k^{(t+\frac{1}{2})}$
6        update local variable $\mathbf{x}_{[k]}^{(t+1)} := \mathbf{x}_{[k]}^{(t)} + \gamma \Delta \mathbf{x}_{[k]}$
7        compute update of local estimate $\Delta \mathbf{v}_k := \mathbf{A}_{[k]} \Delta \mathbf{x}_{[k]}$
8        $\mathbf{v}_k^{(t+1)} := \mathbf{v}_k^{(t+\frac{1}{2})} + \gamma K \Delta \mathbf{v}_k$
9      **end**
10   **end**

---

federated optimization algorithms (like FedAvg in [McMahan et al., 2017]) are still close to the centralized setting. In contrast, our work provides a fully decentralized alternative algorithm for federated learning with generalized linear models.

## 2   The decentralized algorithm: COLA

The COLA framework is summarized in Algorithm 1. For a given input problem we map it to either of the (A) or (B) formulation, and define the locally stored dataset $\mathbf{A}_{[k]}$ and local part of the weight vector $\mathbf{x}_{[k]}$ in node $k$ accordingly. While $\mathbf{v} = \mathbf{A}\mathbf{x}$ is the shared state being communicated in COCOA, this is generally unknown to a node in the fully decentralized setting. Instead, we maintain $\mathbf{v}_k$, a local estimate of $\mathbf{v}$ in node $k$, and use it as a surrogate in the algorithm.

**New data-local quadratic subproblems.** During a computation step, node $k$ locally solves the following minimization problem

$$\min_{\Delta \mathbf{x}_{[k]} \in \mathbb{R}^n} \mathscr{G}_k^{\sigma'}(\Delta \mathbf{x}_{[k]}; \mathbf{v}_k, \mathbf{x}_{[k]}), \tag{1}$$

where

$$\begin{aligned}\mathscr{G}_k^{\sigma'}(\Delta \mathbf{x}_{[k]}; \mathbf{v}_k, \mathbf{x}_{[k]}) := {} & \tfrac{1}{K} f(\mathbf{v}_k) + \nabla f(\mathbf{v}_k)^\top \mathbf{A}_{[k]} \Delta \mathbf{x}_{[k]} \\ & + \tfrac{\sigma'}{2\tau} \left\| \mathbf{A}_{[k]} \Delta \mathbf{x}_{[k]} \right\|^2 + \sum_{i \in \mathcal{P}_k} g_i(x_i + (\Delta \mathbf{x}_{[k]})_i).\end{aligned} \tag{2}$$

Crucially, this subproblem only depends on the local data $\mathbf{A}_{[k]}$, and local vectors $\mathbf{v}_l$ from the neighborhood of the current node $k$. In contrast, in COCOA [Smith et al., 2018] the subproblem is defined in terms of a global aggregated shared vector $\mathbf{v}_c := \mathbf{A}\mathbf{x} \in \mathbb{R}^d$, which is not available in the decentralized setting.[2] The aggregation parameter $\gamma \in [0, 1]$ does not need to be tuned; in fact, we use the default $\gamma := 1$ throughout the paper, see [Ma et al., 2015] for a discussion. Once $\gamma$ is settled, a safe choice of the subproblem relaxation parameter $\sigma'$ is given as $\sigma' := \gamma K$. $\sigma'$ can be additionally tightened using an improved Hessian subproblem (Appendix E.3).

**Algorithm description.** At time $t$ on node $k$, $\mathbf{v}_k^{(t+\frac{1}{2})}$ is a local estimate of the shared variable after a communication step (i.e. gossip mixing). The local subproblem (1) based on this estimate is solved

$$\min_{\Delta \mathbf{x}_{[k]} \in \mathbb{R}^n} \tfrac{1}{K} f(\mathbf{A}(\mathbf{x} + K\Delta \mathbf{x}_{[k]})) + \sum_{i \in \mathcal{P}_k} g_i(x_i + (\Delta \mathbf{x}_{[k]})_i), \tag{3}$$

which is a scaled block-coordinate update of block $k$ of the original objective (A). This assumes that we have consensus $\mathbf{v}_k \equiv \mathbf{A}\mathbf{x} \; \forall \; k$. For *quadratic* objectives (i.e. when $f \equiv \|.\|_2^2$ and $\mathbf{A}$ describes the quadratic), the equality of the formulations (2) and (3) holds. Furthermore, by convexity of $f$, the sum of (3) is an upper bound on the centralized updates $f(\mathbf{x} + \Delta \mathbf{x}) + g(\mathbf{x} + \Delta \mathbf{x})$. Both inequalities quantify the overhead of the distributed algorithm over the centralized version, see also [Yang, 2013, Ma et al., 2015, Smith et al., 2018] for the non-decentralized case.

and yields $\Delta\mathbf{x}_{[k]}$. Then we calculate $\Delta\mathbf{v}_k := \mathbf{A}_{[k]}\Delta\mathbf{x}_{[k]}$, and update the local shared vector $\mathbf{v}_k^{(t+1)}$. We allow the local subproblem to be solved approximately:

**Assumption 1** ($\Theta$-approximation solution). *Let $\Theta \in [0,1]$ be the relative accuracy of the local solver (potentially randomized), in the sense of returning an approximate solution $\Delta\mathbf{x}_{[k]}$ at each step $t$, s.t.*

$$\frac{\mathbb{E}[\mathscr{G}_k^{\sigma'}(\Delta\mathbf{x}_{[k]}; \mathbf{v}_k, \mathbf{x}_{[k]}) - \mathscr{G}_k^{\sigma'}(\Delta\mathbf{x}_{[k]}^\star; \mathbf{v}_k, \mathbf{x}_{[k]})]}{\mathscr{G}_k^{\sigma'}(\ \mathbf{0}\ \ ; \mathbf{v}_k, \mathbf{x}_{[k]}) - \mathscr{G}_k^{\sigma'}(\Delta\mathbf{x}_{[k]}^\star; \mathbf{v}_k, \mathbf{x}_{[k]})} \leq \Theta,$$

*where $\Delta\mathbf{x}_{[k]}^\star \in \arg\min_{\Delta\mathbf{x}\in\mathbb{R}^n} \mathscr{G}_k^{\sigma'}(\Delta\mathbf{x}_{[k]}; \mathbf{v}_k, \mathbf{x}_{[k]})$, for each $k \in [K]$.*

**Elasticity to network size, compute resources and changing data—and fault tolerance.** Real-world communication networks are not homogeneous and static, but greatly vary in availability, computation, communication and storage capacity. Also, the training data is subject to changes. While these issues impose significant challenges for most existing distributed training algorithms, we hereby show that COLA offers adaptivity to such dynamic and heterogenous scenarios.

Scalability and elasticity in terms of availability and computational capacity can be modelled by a node-specific local accuracy parameter $\Theta_k$ in Assumption 1, as proposed by [Smith et al., 2017]. The more resources node $k$ has, the more accurate (smaller) $\Theta_k$ we can use. The same mechanism also allows dealing with fault tolerance and stragglers, which is crucial e.g. on a network of personal devices. More specifically, when a new node $k$ joins the network, its $\mathbf{x}_{[k]}$ variables are initialized to $\mathbf{0}$; when node $k$ leaves, its $\mathbf{x}_{[k]}$ is frozen, and its subproblem is not touched anymore (i.e. $\Theta_k = 1$). Using the same approach, we can adapt to dynamic changes in the dataset—such as additions and removal of local data columns—by adjusting the size of the local weight vector accordingly. Unlike gradient-based methods and ADMM, COLA does not require parameter tuning to converge, increasing resilience to drastic changes.

**Extension to improved second-order subproblems.** In the centralized setting, it has recently been shown that the Hessian information of $f$ can be properly utilized to define improved local subproblems [Lee and Chang, 2017, Dünner et al., 2018]. Similar techniques can be applied to COLA as well, details on which are left in Appendix E.

**Extension to time-varying graphs.** Similar to scalability and elasticity, it is also straightforward to extend COLA to a time varying graph under proper assumptions. If we use the time-varying model in [Nedic et al., 2017, Assumption 1], where an undirected graph is connected with $B$ gossip steps, then changing COLA to perform $B$ communication steps and one computation step per round still guarantees convergence. Details of this setup are provided in Appendix E.

## 3   On the convergence of COLA

In this section we present a convergence analysis of the proposed decentralized algorithm COLA for both general convex and strongly convex objectives. In order to capture the evolution of COLA, we reformulate the original problem (A) by incorporating both $\mathbf{x}$ and local estimates $\{\mathbf{v}_k\}_{k=1}^K$

$$\min_{\mathbf{x}, \{\mathbf{v}_k\}_{k=1}^K} \mathcal{H}_A(\mathbf{x}, \{\mathbf{v}_k\}_{k=1}^K) := \tfrac{1}{K}\sum_{k=1}^K f(\mathbf{v}_k) + g(\mathbf{x}), \tag{DA}$$
$$\text{such that}\qquad \mathbf{v}_k = \mathbf{A}\mathbf{x},\ \ k = 1, ..., K.$$

While the consensus is not always satisfied during Algorithm 1, the following relations between the decentralized objective and the original one (A) always hold. All proofs are deferred to Appendix C.

**Lemma 1.** *Let $\{\mathbf{v}_k\}$ and $\mathbf{x}$ be the iterates generated during the execution of Algorithm 1. At any timestep, it holds that*

$$\tfrac{1}{K}\sum_{k=1}^K \mathbf{v}_k = \mathbf{A}\mathbf{x}, \tag{4}$$
$$F_A(\mathbf{x}) \leq \mathcal{H}_A(\mathbf{x}, \{\mathbf{v}_k\}_{k=1}^K) \leq F_A(\mathbf{x}) + \tfrac{1}{2\tau K}\sum_{k=1}^K \|\mathbf{v}_k - \mathbf{A}\mathbf{x}\|^2. \tag{5}$$

The dual problem and duality gap of the decentralized objective (DA) are given in Lemma 2.

**Lemma 2** (Decentralized Dual Function and Duality Gap). *The Lagrangian dual of the decentralized formation (DA) is*

$$\min_{\{\mathbf{w}_k\}_{k=1}^K} \mathcal{H}_B(\{\mathbf{w}_k\}_{k=1}^K) := \tfrac{1}{K}\sum_{k=1}^K f^*(\mathbf{w}_k) + \sum_{i=1}^n g_i^*\left(-\mathbf{A}_i^\top(\tfrac{1}{K}\sum_{k=1}^K \mathbf{w}_k)\right). \tag{DB}$$

Given primal variables $\{\mathbf{x}, \{\mathbf{v}_k\}_{k=1}^K\}$ and dual variables $\{\mathbf{w}_k\}_{k=1}^K$, the duality gap is:

$$G_{\mathcal{H}}(\mathbf{x}, \{\mathbf{v}_k\}_{k=1}^K, \{\mathbf{w}_k\}_{k=1}^K) := \tfrac{1}{K}\sum_k (f(\mathbf{v}_k)+f^*(\mathbf{w}_k)) + g(\mathbf{x}) + \sum_{i=1}^n g_i^*\left(-\tfrac{1}{K}\sum_k \mathbf{A}_i^\top \mathbf{w}_k\right). \quad (6)$$

If the dual variables are fixed to the optimality condition $\mathbf{w}_k = \nabla f(\mathbf{v}_k)$, then the dual variables can be omitted in the argument list of duality gap, namely $G_{\mathcal{H}}(\mathbf{x}, \{\mathbf{v}_k\}_{k=1}^K)$. Note that the decentralized duality gap generalizes the duality gap of COCOA: when consensus is ensured, i.e., $\mathbf{v}_k \equiv \mathbf{A}\mathbf{x}$ and $\mathbf{w}_k \equiv \nabla f(\mathbf{A}\mathbf{x})$, the decentralized duality gap recovers that of COCOA.

### 3.1 Linear rate for strongly convex objectives

We use the following data-dependent quantities in our main theorems

$$\sigma_k := \max_{\mathbf{x}_{[k]} \in \mathbb{R}^n} \left\|\mathbf{A}_{[k]}\mathbf{x}_{[k]}\right\|^2 / \|\mathbf{x}_{[k]}\|^2, \ \sigma_{\max} = \max_{k=1,\ldots,K} \sigma_k, \ \sigma := \sum_{k=1}^K \sigma_k n_k. \quad (7)$$

If $\{g_i\}$ are strongly convex, COLA achieves the following linear rate of convergence.

**Theorem 1** (Strongly Convex $g_i$). *Consider Algorithm 1 with $\gamma := 1$ and let $\Theta$ be the quality of the local solver in Assumption 1. Let $g_i$ be $\mu_g$-strongly convex for all $i \in [n]$ and let $f$ be $1/\tau$-smooth. Let $\bar{\sigma}' := (1+\beta)\sigma'$, $\alpha := (1 + \frac{(1-\beta)^2}{36(1+\Theta)\beta})^{-1}$ and $\eta := \gamma(1-\Theta)(1-\alpha)$*

$$s_0 = \frac{\tau\mu_g}{\tau\mu_g + \sigma_{max}\bar{\sigma}'} \in [0,1]. \quad (8)$$

*Then after $T$ iterations of Algorithm 1 with[3]*

$$T \geq \frac{1+\eta s_0}{\eta s_0} \log \frac{\varepsilon_{\mathcal{H}}^{(0)}}{\varepsilon_{\mathcal{H}}},$$

*it holds that $\mathbb{E}\big[\mathcal{H}_A(\mathbf{x}^{(T)}, \{\mathbf{v}_k^{(T)}\}_{k=1}^K) - \mathcal{H}_A(\mathbf{x}^\star, \{\mathbf{v}_k^\star\}_{k=1}^K)\big] \leq \varepsilon_{\mathcal{H}}$. Furthermore, after $T$ iterations with*

$$T \geq \frac{1+\eta s_0}{\eta s_0} \log \left(\frac{1}{\eta s_0} \frac{\varepsilon_{\mathcal{H}}^{(0)}}{\varepsilon_{G_{\mathcal{H}}}},\right)$$

*we have the expected duality gap $\mathbb{E}[G_{\mathcal{H}}(\mathbf{x}^{(T)}, \{\sum_{k=1}^K \mathcal{W}_{kl}\mathbf{v}_l^{(T)}\}_{k=1}^K)] \leq \varepsilon_{G_{\mathcal{H}}}$.*

### 3.2 Sublinear rate for general convex objectives

Models such as sparse logistic regression, Lasso, group Lasso are non-strongly convex. For such models, we show that COLA enjoys a $\mathcal{O}(1/T)$ sublinear rate of convergence for all network topologies with a positive spectral gap.

**Theorem 2** (Non-strongly Convex Case). *Consider Algorithm 1, using a local solver of quality $\Theta$. Let $g_i(\cdot)$ have L-bounded support, and let $f$ be $(1/\tau)$-smooth. Let $\varepsilon_{G_{\mathcal{H}}} > 0$ be the desired duality gap. Then after $T$ iterations where*

$$T \geq T_0 + \max\left\{\left\lceil\tfrac{1}{\eta}\right\rceil, \tfrac{4L^2\sigma\bar{\sigma}'}{\tau\varepsilon_{G_{\mathcal{H}}}\eta}\right\}, \qquad T_0 \geq t_0 + \left[\tfrac{2}{\eta}\left(\tfrac{8L^2\sigma\bar{\sigma}'}{\tau\varepsilon_{G_{\mathcal{H}}}} - 1\right)\right]_+$$

$$t_0 \geq \max\left\{0, \left\lceil\tfrac{1+\eta}{\eta}\log\tfrac{2\tau(\mathcal{H}_A(\mathbf{x}^{(0)}, \{\mathbf{v}_i^{(0)}\}) - \mathcal{H}_A(\mathbf{x}^\star, \{\mathbf{v}^\star\}))}{4L^2\sigma\bar{\sigma}'}\right\rceil\right\}$$

*and $\bar{\sigma}' := (1+\beta)\sigma'$, $\alpha := (1 + \frac{(1-\beta)^2}{36(1+\Theta)\beta})^{-1}$ and $\eta := \gamma(1-\Theta)(1-\alpha)$. We have that the expected duality gap satisfies*

$$\mathbb{E}\big[G_{\mathcal{H}}(\bar{\mathbf{x}}, \{\bar{\mathbf{v}}_k\}_{k=1}^K, \{\bar{\mathbf{w}}_k\}_{k=1}^K)\big] \leq \varepsilon_{G_{\mathcal{H}}}$$

*at the averaged iterate $\bar{\mathbf{x}} := \frac{1}{T-T_0}\sum_{t=T_0+1}^{T-1} \mathbf{x}^{(t)}$, and $\mathbf{v}_k' := \sum_{l=1}^K \mathcal{W}_{kl}\mathbf{v}_l$ and $\bar{\mathbf{v}}_k := \frac{1}{T-T_0}\sum_{t=T_0+1}^{T-1} (\mathbf{v}_k')^{(t)}$ and $\bar{\mathbf{w}}_k := \frac{1}{T-T_0}\sum_{t=T_0+1}^{T-1} \nabla f((\mathbf{v}_k')^{(t)})$.*

Note that the assumption of bounded support for the $g_i$ functions is not restrictive in the general convex case, as discussed e.g. in [Dünner et al., 2016].

### 3.3 Local certificates for global accuracy

Accuracy certificates for the training error are very useful for practitioners to diagnose the learning progress. In the centralized setting, the duality gap serves as such a certificate, and is available as a stopping criterion on the master node. In the decentralized setting of our interest, this is more challenging as consensus is not guaranteed. Nevertheless, we show in the following Proposition 1 that certificates for the decentralized objective (DA) can be computed from local quantities:

**Proposition 1** (Local Certificates). *Assume $g_i$ has $L$-bounded support, and let $\mathcal{N}_k := \{j : \mathcal{W}_{jk} > 0\}$ be the set of nodes accessible to node $k$. Then for any given $\varepsilon > 0$, we have*

$$G_{\mathcal{H}}(\mathbf{x}; \{\mathbf{v}_k\}_{k=1}^K) \leq \varepsilon,$$

*if for all $k = 1, \dots, K$ the following two local conditions are satisfied:*

$$\mathbf{v}_k^\top \nabla f(\mathbf{v}_k) + \sum_{i \in \mathcal{P}_k} \left(g_i(\mathbf{x}_i) + g_i^*(-\mathbf{A}_i^\top \nabla f(\mathbf{v}_k))\right) \leq \frac{\varepsilon}{2K} \tag{9}$$

$$\left\| \nabla f(\mathbf{v}_k) - \tfrac{1}{|\mathcal{N}_k|} \sum_{j \in \mathcal{N}_k} \nabla f(\mathbf{v}_j) \right\|_2 \leq \left(\sum_{k=1}^K n_k^2 \sigma_k\right)^{-1/2} \frac{1-\beta}{2L\sqrt{K}} \varepsilon, \tag{10}$$

The local conditions (9) and (10) have a clear interpretation. The first one ensures the duality gap of the local subproblem given by $\mathbf{v}_k$ as on the left hand side of (9) is small. The second condition (10) guarantees that consensus violation is bounded, by ensuring that the gradient of each node is similar to its neighborhood nodes.

**Remark 1.** *The resulting certificate from Proposition 1 is local, in the sense that no global vector aggregations are needed to compute it. For a certificate on the global objective, the boolean flag of each local condition (9) and (10) being satisfied or not needs to be shared with all nodes, but this requires extremely little communication. Exact values of the parameters $\beta$ and $\sum_{k=1}^K n_k^2 \sigma_k$ are not required to be known, and any valid upper bound can be used instead. We can use the local certificates to avoid unnecessary work on local problems which are already optimized, as well as to continuously quantify how newly arriving local data has to be re-optimized in the case of online training. The local certificates can also be used to quantify the contribution of newly joining or departing nodes, which is particularly useful in the elastic scenario described above.*

## 4 Experimental results

Here we illustrate the advantages of COLA in three respects: firstly we investigate the application in different network topologies and with varying subproblem quality $\Theta$; secondly, we compare COLA with state-of-the-art decentralized baselines: ①, DIGing [Nedic et al., 2017], which generalizes the gradient-tracking technique of the EXTRA algorithm [Shi et al., 2015], and ②, Decentralized ADMM (aka. consensus ADMM), which extends the classical ADMM (Alternating Direction Method of

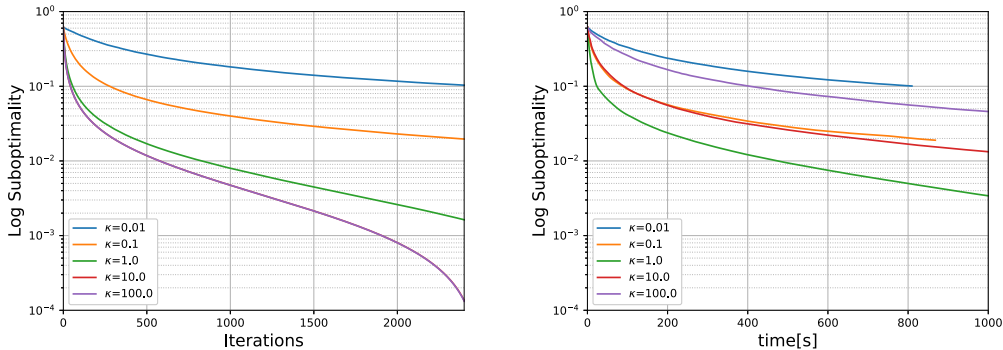

Figure 1: Suboptimality for solving Lasso ($\lambda=10^{-6}$) for the RCV1 dataset on a ring of 16 nodes. We illustrate the performance of COLA: a) number of iterations; b) time. $\kappa$ here denotes the number of local data passes per communication round.

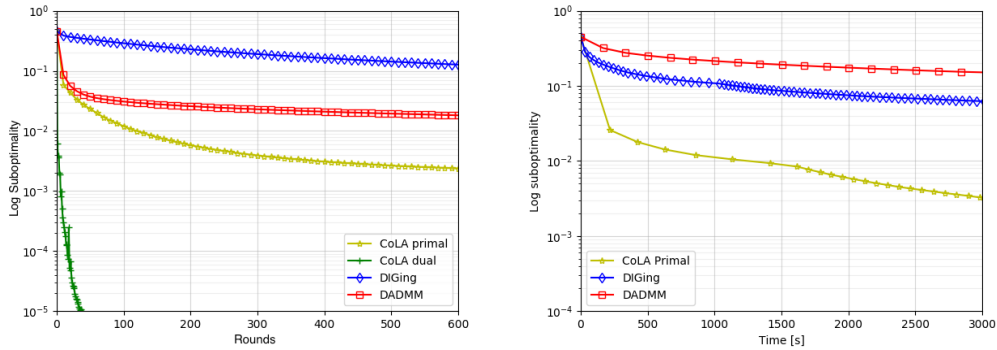

Figure 2: Convergence of CoLa for solving problems on a ring of $K{=}16$ nodes. Left) Ridge regression on URL reputation dataset ($\lambda{=}10^{-4}$); Right) Lasso on webspam dataset ($\lambda{=}10^{-5}$).

Multipliers) method [Boyd et al., 2011] to the decentralized setting [Shi et al., 2014, Wei and Ozdaglar, 2013]; Finally, we show that CoLa works in the challenging unreliable network environment where each node has a certain chance to drop out of the network.

We implement all algorithms in PyTorch with MPI backend. The decentralized network topology is simulated by running one thread per graph node, on a $2{\times}12$ core Intel Xeon CPU E5-2680 v3 server with 256 GB RAM. Table 1 describes the datasets[4] used in the experiments. For Lasso, the columns of $\mathbf{A}$ are features. For ridge regression, the columns are features and samples for CoLa primal and CoLa dual, respectively. The order of columns is shuffled once before being distributed across the nodes. Due to space limit, details on the experimental configurations are included in Appendix D.

**Effect of approximation quality $\Theta$.** We study the convergence behavior in terms of the approximation quality $\Theta$. Here, $\Theta$ is controlled by the number of data passes $\kappa$ on subproblem (1) per node. Figure 1 shows that increasing $\kappa$ always results in less number of iterations (less communication rounds) for CoLa. However, given a fixed network bandwidth, it leads to a clear

Table 1: Datasets Used for Empirical Study

| Dataset | #Training | #Features | Sparsity |
|---|---|---|---|
| URL | 2M | 3M | 3.5e-5 |
| Webspam | 350K | 16M | 2.0e-4 |
| Epsilon | 400K | 2K | 1.0 |
| RCV1 Binary | 677K | 47K | 1.6e-3 |

trade-off for the overall wall-clock time, showing the cost of both communication and computation. Larger $\kappa$ leads to less communication rounds, however, it also takes more time to solve subproblems. The observations suggest that one can adjust $\Theta$ for each node to handle system heterogeneity, as what we have discussed at the end of Section 2.

**Effect of graph topology.** Fixing $K{=}16$, we test the performance of CoLa on 5 different topologies: ring, 2-connected cycle, 3-connected cycle, 2D grid and complete graph. The mixing matrix $\mathcal{W}$ is given by Metropolis weights for all test cases (details in Appendix B). Convergence curves are plotted in Figure 3. One can observe that for all topologies, CoLa converges monotonically and especially when all nodes in the network are equal, smaller $\beta$ leads to a faster convergence rate. This is consistent with the intuition that $1 - \beta$ measures the connectivity level of the topology.

**Superior performance compared to baselines.** We compare CoLa with DIGing and D-ADMM for strongly and general convex problems. For general convex objectives, we use Lasso regression with $\lambda = 10^{-4}$ on the webspam dataset; for the strongly convex objective, we use Ridge regression with $\lambda = 10^{-5}$ on the URL reputation dataset. For Ridge regression, we can map CoLa to both primal and dual problems. Figure 2 traces the results on log-suboptimality. One can observe that for both generally and strongly convex objectives, CoLa significantly outperforms DIGing and decentralized ADMM in terms of number of communication rounds and computation time. While DIGing and D-ADMM need parameter tuning to ensure convergence and efficiency, CoLa is much easier to deploy as it is parameter free. Additionally, convergence guarantees of ADMM relies on exact subproblem solvers, whereas inexact solver is allowed for CoLa.

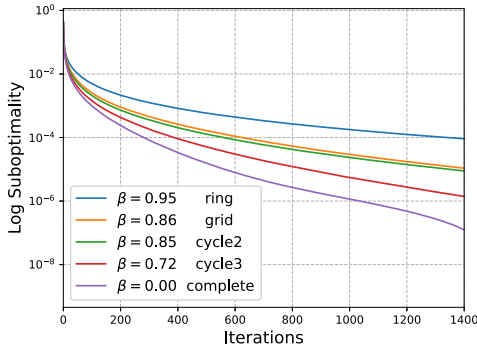

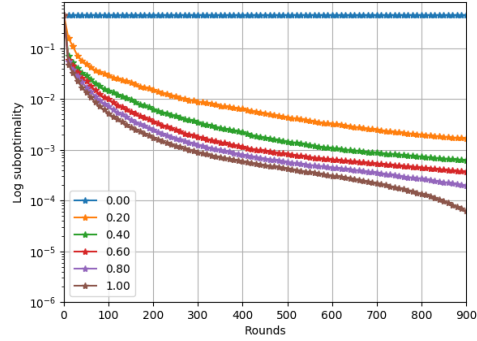

Figure 3: Performance comparison of CoLA on different topologies. Solving Lasso regression ($\lambda=10^{-6}$) for RCV1 dataset with 16 nodes.

Figure 4: Performance of CoLA when nodes have $p$ chance of staying in the network on the URL dataset ($\lambda=10^{-4}$). Freezing $\mathbf{x}_{[k]}$ when node $k$ leaves the network.

**Fault tolerance to unreliable nodes.** Assume each node of a network only has a chance of $p$ to participate in each round. If a new node $k$ joins the network, then local variables are initialized as $\mathbf{x}_{[k]} = 0$; if node $k$ leaves the network, then $\mathbf{x}_{[k]}$ will be frozen with $\Theta_k = 1$. All remaining nodes dynamically adjust their weights to maintain the doubly stochastic property of $\mathcal{W}$. We run CoLA on such unreliable networks of different $p$s and show the results in Figure 4. First, one can observe that for all $p > 0$ the suboptimality decreases monotonically as CoLA progresses. It is also clear from the result that a smaller dropout rate (a larger $p$) leads to a faster convergence of CoLA.

## 5  Discussion and conclusions

In this work we have studied training generalized linear models in the fully decentralized setting. We proposed a communication-efficient decentralized framework, termed CoLA, which is free of parameter tuning. We proved that it has a sublinear rate of convergence for general convex problems, allowing e.g. L1 regularizers, and has a linear rate of convergence for strongly convex objectives. Our scheme offers primal-dual certificates which are useful in the decentralized setting. We demonstrated that CoLA offers full adaptivity to heterogenous distributed systems on arbitrary network topologies, and is adaptive to changes in network size and data, and offers fault tolerance and elasticity. Future research directions include improving subproblems, as well as extension to the network topology with directed graphs, as well as recent communication compression schemes [Stich et al., 2018].

**Acknowledgments.** We thank Prof. Bharat K. Bhargava for fruitful discussions. We acknowledge funding from SNSF grant 200021_175796, Microsoft Research JRC project 'Coltrain', as well as a Google Focused Research Award.

## Footnotes

[2]*Subproblem interpretation:* Note that for the special case of $\gamma := 1$, $\sigma' := K$, by smoothness of $f$, our subproblem in (2) is an upper bound on

[3] $\varepsilon_{\mathcal{H}}^{(0)} := \mathcal{H}_A(\mathbf{x}^{(0)}, \{\mathbf{v}_k^{(0)}\}_{k=1}^K) - \mathcal{H}_A(\mathbf{x}^\star, \{\mathbf{v}_k^\star\}_{k=1}^K)$ is the initial suboptimality.

[4]https://www.csie.ntu.edu.tw/~cjlin/libsvmtools/datasets/

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
