[Supplementary Material · supplementary.pdf]

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

# Appendix

## A  Definitions

**Definition 1** (*L*-Lipschitz continuity). *A function $h : \mathbb{R}^n \to \mathbb{R}$ is L-Lipschitz continuous if $\forall \, \mathbf{u}, \mathbf{v} \in \mathbb{R}^n$, it holds that*
$$|h(\mathbf{u}) - h(\mathbf{v})| \leq L\|\mathbf{u} - \mathbf{v}\|.$$

**Definition 2** ($1/\tau$-Smoothness). *A differentiable function $f : \mathbb{R}^n \to \mathbb{R}$ is $1/\tau$-smooth if its gradient is $1/\tau$-Lipschitz continuous, or equivalently, $\forall \, \mathbf{u}, \mathbf{v}$ it holds*
$$f(\mathbf{u}) \leq f(\mathbf{v}) + \langle \nabla f(\mathbf{v}), \mathbf{u} - \mathbf{v} \rangle + \tfrac{1}{2\tau}\|\mathbf{u} - \mathbf{v}\|^2. \tag{11}$$

**Definition 3** (*L*-Bounded support). *The function $g : \mathbb{R}^n \to \mathbb{R} \cup \{+\infty\}$ has L-bounded support if it holds $g(\mathbf{u}) < +\infty \Rightarrow \|\mathbf{u}\| \leq L$.*

**Definition 4** ($\mu$-Strong convexity). *A function $h : \mathbb{R}^n \to \mathbb{R}$ is $\mu$-strongly convex for $\mu \geq 0$ if $\forall \, \mathbf{u}, \mathbf{v}$, it holds $h(\mathbf{u}) \geq h(\mathbf{v}) + \langle \mathbf{s}, \mathbf{u} - \mathbf{v} \rangle + \tfrac{\mu}{2}\|\mathbf{u} - \mathbf{v}\|^2$, for any $\mathbf{s} \in \partial h(\mathbf{v})$, where $\partial h(\mathbf{v})$ is the subdifferential of $h$ at $\mathbf{v}$.*

**Lemma 3** (Duality between Lipschitzness and L-Bounded Support). *A generalization of [Rockafellar, 2015, Corollary 13.3.3]. Given a proper convex function $g$ it holds that $g$ has L-bounded support w.r.t. the norm $\|.\|$ if and only if $g^*$ is L-Lipschitz w.r.t. the dual norm $\|.\|_*$.*

## B  Graph topology

Let $\mathcal{E}$ be the set of edges of a graph. For time-invariant undirected graph the mixing matrix should satisfy the following properties:

1. (*Double stochasticity*) $\mathcal{W}\mathbf{1} = \mathbf{1}, \mathbf{1}^\top\mathcal{W} = \mathbf{1}^\top$;
2. (*Symmetrization*) For all $i, j$, $\mathcal{W}_{ij} = \mathcal{W}_{ji}$;
3. (*Edge utilization*) If $(i,j) \in \mathcal{E}$, then $\mathcal{W}_{ij} > 0$; otherwise $\mathcal{W}_{ij} = 0$.

A desired mixing matrix can be constructed using Metropolis-Hastings weights [Hastings, 1970]:

$$\mathcal{W}_{ij} = \begin{cases} 1/(1 + \max\{d_i, d_j\}), & \text{if } (i,j) \in \mathcal{E} \\ 0, & \text{if } (i,j) \notin \mathcal{E} \text{ and } j \neq i \\ 1 - \sum_{l \in \mathcal{N}_i} \mathcal{W}_{il}, & \text{if } j = i, \end{cases}$$

where $d_i = |\mathcal{N}_i|$ is the degree of node $i$.

## C  Proofs

This section consists of three parts. Tools and observations are provided in Appendix C.1; The main lemmas for the convergence analysis are proved in Appendix C.2 ; The main theorems and implications are proved in Appendix C.3.

In some circumstances, it is convenient to use notations of array of stack column vectors. For example, one can stack local estimates $\mathbf{v}_k$ to matrix $\mathbf{V} := [\mathbf{v}_1; \cdots; \mathbf{v}_K]$, $\Delta\mathbf{V} = [\Delta\mathbf{v}_1; \cdots; \Delta\mathbf{v}_K]$. The consensus vector $\mathbf{v}_c$ is repeated $K$ times which will be stacked similarly: $\mathbf{V}_c := \mathbf{A}\mathbf{x}\mathbf{1}_K^\top = \mathbf{V}\mathbf{E}$ where $\mathbf{E} = \tfrac{1}{K}\mathbf{1}_K\mathbf{1}_K^\top$. The consensus violation under the two notations is written as

$$\|\mathbf{V} - \mathbf{V}_c\|_F^2 = \sum_{k=1}^K \|\mathbf{v}_k - \mathbf{A}\mathbf{x}\|_2^2.$$

Then Step 8 in COLA is equivalent to

$$\mathbf{V}^{(t+1)} = \mathbf{V}^{(t)}\mathcal{W} + \gamma K \Delta\mathbf{V}^{(t)} \tag{12}$$

Besides, we also adopt following notations in the proof when there is no ambiguity: $\mathbf{v}'_k := \sum_{l=1}^K \mathcal{W}_{kl}\mathbf{v}_l$, $\mathbf{g}_k := \nabla f(\mathbf{v}_k)$, $\mathbf{g}'_k := \nabla f(\mathbf{v}'_k)$ and $\bar{\mathbf{g}} := \tfrac{1}{K}\sum_{k=1}^K \mathbf{g}_k$. For the decentralized duality

gap $G_{\mathcal{H}}(\mathbf{x}, \{\mathbf{v}_k\}_{k=1}^K, \{\mathbf{w}_k\}_{k=1}^K)$, when $\mathbf{w}_k = \nabla f(\mathbf{v}_k)$, we simplify $G_{\mathcal{H}}(\mathbf{x}, \{\mathbf{v}_k\}_{k=1}^K, \{\mathbf{w}_k\}_{k=1}^K)$ to be $G_{\mathcal{H}}(\mathbf{x}, \{\mathbf{v}_k\}_{k=1}^K)$ in the sequel.

On a high level, we prove the convergence rates by bounding per-iteration reduction $\mathbb{E}[\mathcal{H}_A^{(t)} - \mathcal{H}_A^{(t+1)}]$ using decentralized duality gap and other related terms, then try to obtain the final rates by properly using specific properties of the objectives.

However, the specific analysis of the new fully decentralized algorithm COLA poses many new challenges, and we propose significantly new proof techniques in the analysis. Specifically, i) we introduce the decentralized duality gap, which is suited for the decentralized algorithm COLA; ii) consensus violation is the usually challenging part in analyzing decentralized algorithms. Unlike using uniform bounds for consensus violations, e.g., [Yuan et al., 2016], we properly combine the consensus violation term and the objective decrease term (c.f. Lemmas 6 and 8), thus reaching arguably tight convergence bounds for both the consensus violation term and the objective.

### C.1 Observations and properties

In this subsection we introduce basic lemmas. Lemma 1 establishes the relation between $\{\mathbf{v}_k\}_{k=1}^K$ and $\mathbf{v}_c$ and bounds $F_A(\mathbf{x})$ using $\mathcal{H}_A(\mathbf{x})$ and the consensus violation.

**Lemma 1.** *Let $\{\mathbf{v}_k\}$ and $\mathbf{x}$ be the iterates generated during the execution of Algorithm 1. At any timestep, it holds that*

$$\frac{1}{K} \sum_{k=1}^K \mathbf{v}_k = \mathbf{A}\mathbf{x}, \tag{4}$$

$$F_A(\mathbf{x}) \le \mathcal{H}_A(\mathbf{x}, \{\mathbf{v}_k\}_{k=1}^K) \le F_A(\mathbf{x}) + \frac{1}{2\tau K} \sum_{k=1}^K \|\mathbf{v}_k - \mathbf{A}\mathbf{x}\|^2. \tag{5}$$

*Proof of Lemma 1.* Let $\widetilde{\mathbf{v}} := \frac{1}{K} \sum_{k=1}^K \mathbf{v}_k$. Using the doubly stochastic property of the matrix $\mathcal{W}$

$$\widetilde{\mathbf{v}}^{(t+1)} = \frac{1}{K} \sum_{k=1}^K \mathbf{v}_k^{(t+1)} = \frac{1}{K} \sum_{k=1}^K \left( \sum_{l=1}^K \mathcal{W}_{kl} \mathbf{v}_l^{(t)} + \gamma K \Delta \mathbf{v}_k^{(t)} \right)$$

$$= \frac{1}{K} \sum_{l=1}^K \mathbf{v}_l^{(t)} + \gamma \sum_{k=1}^K \Delta \mathbf{v}_k^{(t)} = \widetilde{\mathbf{v}}^{(t)} + \gamma \sum_{k=1}^K \Delta \mathbf{v}_k^{(t)}$$

On the other hand, $\mathbf{v}_c^{(t)} := \mathbf{A}\mathbf{x}^{(t)}$ is updated based on all changes of local variables $\{\mathbf{x}_{[k]}\}_{k=1}^K$

$$\mathbf{v}_c^{(t+1)} = \mathbf{v}_c^{(t)} + \gamma \sum_{k=1}^K \Delta \mathbf{v}_k^{(t)}.$$

Since $\widetilde{\mathbf{v}}^{(0)} = \mathbf{v}_c^{(0)}$, we can conclude that $\widetilde{\mathbf{v}}^{(t)} = \mathbf{v}_c^{(t)} \ \forall \, t$. From convexity of $f$ we know

$$F_A(\mathbf{x}) = f(\mathbf{v}_c) + g(\mathbf{x}) = f\left( \frac{1}{K} \sum_{k=1}^K \mathbf{v}_k \right) + g(\mathbf{x}) \le \frac{1}{K} \sum_{k=1}^K f(\mathbf{v}_k) + g(\mathbf{x}) = \mathcal{H}(\mathbf{x})$$

Using $1/\tau$-smoothness of $f$ gives

$$\mathcal{H}_A(\mathbf{x}) = \frac{1}{K} \sum_{k=1}^K f(\mathbf{v}_k) + g(\mathbf{x})$$

$$\le \frac{1}{K} \sum_{k=1}^K \left( f(\mathbf{v}_c) + \nabla f(\mathbf{v}_c)^\top (\mathbf{v}_k - \mathbf{v}_c) + \frac{1}{2\tau} \|\mathbf{v}_k - \mathbf{v}_c\|^2 \right) + g(\mathbf{x})$$

$$= F_A(\mathbf{x}) + \frac{1}{2\tau K} \sum_{k=1}^K \|\mathbf{v}_k - \mathbf{v}_c\|^2.$$

$\square$

The following lemma introduces the dual problem and the duality gap of (DA).

**Lemma 2** (Decentralized Dual Function and Duality Gap). *The Lagrangian dual of the decentralized formation* (DA) *is*

$$\min_{\{\mathbf{w}_k\}_{k=1}^K} \mathcal{H}_B(\{\mathbf{w}_k\}_{k=1}^K) := \tfrac{1}{K} \sum_{k=1}^K f^*(\mathbf{w}_k) + \sum_{i=1}^n g_i^* \left( -\mathbf{A}_i^\top \left( \tfrac{1}{K} \sum_{k=1}^K \mathbf{w}_k \right) \right). \quad \text{(DB)}$$

*Given primal variables* $\{\mathbf{x}, \{\mathbf{v}_k\}_{k=1}^K\}$ *and dual variables* $\{\mathbf{w}_k\}_{k=1}^K$, *the duality gap is:*

$$G_{\mathcal{H}}(\mathbf{x}, \{\mathbf{v}_k\}_{k=1}^K, \{\mathbf{w}_k\}_{k=1}^K) := \tfrac{1}{K} \sum_k (f(\mathbf{v}_k) + f^*(\mathbf{w}_k)) + g(\mathbf{x}) + \sum_{i=1}^n g_i^* \left( -\tfrac{1}{K} \sum_k \mathbf{A}_i^\top \mathbf{w}_k \right). \quad \text{(6)}$$

*Proof.* Let $\boldsymbol{\lambda}_k$ be the Lagrangian multiplier for the constraint $\mathbf{v}_k = \mathbf{A}\mathbf{x}$, the Lagrangian function is

$$L(\mathbf{x}, \{\mathbf{v}_k\}_{k=1}^K, \{\boldsymbol{\lambda}_k\}_{k=1}^K) = \frac{1}{K} \sum_{k=1}^K f(\mathbf{v}_k) + \sum_{i=1}^n g_i(\mathbf{x}_i) + \sum_{k=1}^K \langle \boldsymbol{\lambda}_k, \mathbf{A}\mathbf{x} - \mathbf{v}_k \rangle$$

The dual problem of (DA) follows by taking the infimum with respect to both $\mathbf{x}$ and $\{\mathbf{v}_k\}_{k=1}^K$:

$$\inf_{\mathbf{x}, \{\mathbf{v}_k\}_{k=1}^K} L(\mathbf{x}, \{\mathbf{v}_k\}_{k=1}^K, \{\boldsymbol{\lambda}_k\}_{k=1}^K)$$

$$= \inf_{\mathbf{x}, \{\mathbf{v}_k\}_{k=1}^K} \frac{1}{K} \sum_{k=1}^K f(\mathbf{v}_k) + \sum_{i=1}^n g_i(\mathbf{x}_i) + \sum_{k=1}^K \langle \boldsymbol{\lambda}_k, \mathbf{A}\mathbf{x} - \mathbf{v}_k \rangle$$

$$= \sum_{k=1}^K \inf_{\{\mathbf{v}_k\}_{k=1}^K} \left( \frac{1}{K} f(\mathbf{v}_k) - \langle \boldsymbol{\lambda}_k, \mathbf{v}_k \rangle \right) + \inf_{\mathbf{x}} \left( \sum_{i=1}^n g_i(\mathbf{x}_i) + \sum_{k=1}^K \langle \boldsymbol{\lambda}_k, \mathbf{A}\mathbf{x} \rangle \right)$$

$$= -\sum_{k=1}^K \sup_{\{\mathbf{v}_k\}_{k=1}^K} \left( \langle \boldsymbol{\lambda}_k, \mathbf{v}_k \rangle - \frac{1}{K} f(\mathbf{v}_k) \right) - \sup_{\mathbf{x}} \left( -\sum_{k=1}^K \langle \mathbf{a}_k, \mathbf{A}\mathbf{x} \rangle - \sum_{i=1}^n g_i(\mathbf{x}_i) \right)$$

$$= -\sum_{k=1}^K \frac{1}{K} f^*(K\boldsymbol{\lambda}_k) - \sum_{i=1}^n g_i^* \left( -\sum_{k=1}^K \mathbf{A}_i^\top \boldsymbol{\lambda}_k \right)$$

Let us change variables from $\boldsymbol{\lambda}_k$ to $\mathbf{w}_k$ by setting $\mathbf{w}_k := K\boldsymbol{\lambda}_k$. If written in terms of minimization, the Lagrangian dual of $\mathcal{H}_A$ is

$$\min_{\{\mathbf{w}_k\}_{k=1}^K} \mathcal{H}_B\{\mathbf{w}_k\}_{k=1}^K = \tfrac{1}{K} \sum_{k=1}^K f^*(\mathbf{w}_k) + \sum_{i=1}^n g_i^* \left( -\tfrac{1}{K} \sum_{k=1}^K \mathbf{A}_i^\top \mathbf{w}_k \right) \quad \text{(13)}$$

The optimality condition is that $\mathbf{w}_k = \nabla f(\mathbf{v}_k)$. Now we can see the duality gap is

$$G_{\mathcal{H}} = \mathcal{H}_A + \mathcal{H}_B$$

$$= \frac{1}{K} \sum_{k=1}^K f(\mathbf{v}_k) + \sum_{i=1}^n g_i(x_i) + \frac{1}{K} \sum_{k=1}^K f^*(\mathbf{w}_k) + \sum_{i=1}^n g_i^* \left( -\frac{1}{K} \sum_{k=1}^K \mathbf{A}_i^\top \mathbf{w}_k \right)$$

$\square$

The following lemma correlates the consensus violation with the magnitude of the $\mathbf{v}$ parameter updates $\|\Delta \mathbf{v}_k\|_2^2$.

**Lemma 4.** *The consensus violation during the execution of Algorithm 1 can be bound by*

$$\sum_{k=1}^K \left\| \mathbf{v}_k^{(t+1)} - \mathbf{A}\mathbf{x}^{(t+1)} \right\|_2^2 \le \beta \sum_{k=1}^K \left\| \mathbf{v}_k^{(t)} - \mathbf{A}\mathbf{x}^{(t)} \right\|_2^2 + (1-\beta)c_1(\beta, \gamma, K) \sum_{k=1}^K \left\| \Delta \mathbf{v}_k^{(t)} \right\|_2^2 \quad \text{(14)}$$

*where* $c_1(\beta, \gamma, K) := \gamma^2 K^2 / (1-\beta)^2$.

*Proof.* Consider the norm of consensus violation at time $t+1$ and apply Algo. Step 8

$$\left\| \mathbf{V}^{(t+1)} - \mathbf{V}_c^{(t+1)} \right\|_F^2 = \left\| \mathbf{V}^{(t+1)}(\mathbf{I} - \mathbf{E}) \right\|_F^2 = \left\| (\mathbf{V}^{(t)}\mathcal{W} + \gamma K \Delta \mathbf{V}^{(t)})(\mathbf{I} - \mathbf{E}) \right\|_F^2.$$

Further, use $\mathcal{W}(\mathbf{I} - \mathbf{E}) = (\mathbf{I} - \mathbf{E})(\mathcal{W} - \mathbf{E})$, $\|\mathbf{I} - \mathbf{E}\|_\infty = 1$, and Young's inequality with $\varepsilon_\mathbf{v}$

$$\left\|\mathbf{V}^{(t+1)} - \mathbf{V}_c^{(t+1)}\right\|_F^2 \leq (1 + \varepsilon_\mathbf{v})\left\|\mathbf{V}^{(t)}(\mathbf{I} - \mathbf{E})(\mathcal{W} - \mathbf{E})\right\|_F^2 + (1 + \frac{1}{\varepsilon_\mathbf{v}})\gamma^2 K^2 \left\|\Delta\mathbf{V}^{(t)}\right\|_F^2.$$

Use the spectral property of $\mathcal{W}$ we therefore have:

$$\left\|\mathbf{V}^{(t+1)} - \mathbf{V}_c^{(t+1)}\right\|_F^2 \leq (1 + \varepsilon_\mathbf{v})\beta^2 \left\|\mathbf{V}^{(t)} - \mathbf{V}_c^{(t)}\right\|_F^2 + (1 + \frac{1}{\varepsilon_\mathbf{v}})\gamma^2 K^2 \left\|\Delta\mathbf{V}^{(t)}\right\|_F^2. \tag{15}$$

Recursively apply (15) for $i = 0, \ldots, t-1$ gives

$$\left\|\mathbf{V}^{(t)} - \mathbf{V}_c^{(t)}\right\|_F^2 \leq (1 + \frac{1}{\varepsilon_\mathbf{v}})\gamma^2 K^2 \sum_{i=0}^{t-1}((1 + \varepsilon_\mathbf{v})\beta^2)^{t-1-i} \left\|\Delta\mathbf{V}^{(i)}\right\|_F^2. \tag{16}$$

Consider $\left\|\Delta\mathbf{V}^{(t)}\right\|_F^2$ generated at time $t$, it will be used in (16) from time $t+1, t+2, \ldots$, with coefficients $1, (1 + \varepsilon_\mathbf{v})\beta^2, ((1 + \varepsilon_\mathbf{v})\beta^2)^2, \ldots$. Sum of such coefficients are finite

$$(1 + \frac{1}{\varepsilon_\mathbf{v}})\gamma^2 K^2 \sum_{t=T}^{\infty}((1 + \varepsilon_\mathbf{v})\beta^2)^{t-T} \leq \gamma^2 K^2 \frac{1 + 1/\varepsilon_\mathbf{v}}{1 - (1 + \varepsilon_\mathbf{v})\beta^2} =: c_1(\beta, \gamma, K) \tag{17}$$

where we need $(1 + \varepsilon_\mathbf{v})\beta^2 < 1$. To minimize $c_1(\beta, \gamma, K)$ we can choose $\varepsilon_\mathbf{v} = 1/\beta - 1$

$$c_1(\beta, \gamma, K) = \gamma^2 K^2 / (1 - \beta)^2 \tag{18}$$

Then (15) becomes

$$\sum_{k=1}^{K} \left\|\mathbf{v}_k^{(t+1)} - \mathbf{A}\mathbf{x}^{(t+1)}\right\|_2^2 \leq \beta \sum_{k=1}^{K} \left\|\mathbf{v}_k^{(t)} - \mathbf{A}\mathbf{x}^{(t)}\right\|_2^2 + (1 - \beta)c_1(\beta, \gamma, K) \sum_{k=1}^{K} \left\|\Delta\mathbf{v}_k^{(t)}\right\|_2^2$$

$\square$

**Lemma 5.** *Let $\Delta\mathbf{x}_{[k]}^\star$ and $\Delta\mathbf{x}_{[k]}$ be the exact and $\Theta$-inexact solution of subproblem $\mathscr{G}_k^{\sigma'}(\cdot\,; \mathbf{v}_k, \mathbf{x}_{[k]})$. The change of iterates satisfies the following inequality*

$$\frac{\sigma'}{4\tau} \sum_{k=1}^{K} \left\|\mathbf{A}\Delta\mathbf{x}_{[k]}\right\|_2^2 \leq (1 + \Theta)(\mathcal{H}_A(\mathbf{0}; \{\mathbf{v}_k\}) - \mathcal{H}_A(\Delta\mathbf{x}^\star; \{\mathbf{v}_k\})) \tag{19}$$

*Proof.* First use the Taylor expansion of $\mathscr{G}_k^{\sigma'}(\cdot\,; \mathbf{v}_k, \mathbf{x}_{[k]})$ and the defnition of $\Delta\mathbf{x}_{[k]}^\star$ we have

$$\frac{\sigma'}{2\tau} \left\|\mathbf{A}(\Delta\mathbf{z} - \Delta\mathbf{x}^\star)_{[k]}\right\|_2^2 \leq \mathscr{G}_k^{\sigma'}\left(\Delta\mathbf{z}_{[k]}; \mathbf{v}_k, \mathbf{x}_{[k]}\right) - \mathscr{G}_k^{\sigma'}\left(\Delta\mathbf{x}_{[k]}^\star; \mathbf{v}_k, \mathbf{x}_{[k]}\right) \tag{20}$$

for all $\Delta\mathbf{z}_{[k]} \in \mathbb{R}^n$ and $k = 1, \ldots, K$. Apply (20) with $\Delta\mathbf{z}_{[k]} = \mathbf{0}$ for all $k$ and sum them up yields

$$\frac{\sigma'}{2\tau} \sum_{k=1}^{K} \left\|\mathbf{A}\Delta\mathbf{x}_{[k]}^\star\right\|_2^2 \leq \mathcal{H}_A(\mathbf{0}; \{\mathbf{v}_k\}) - \mathcal{H}_A(\Delta\mathbf{x}^\star; \{\mathbf{v}_k\}) \tag{21}$$

Similarly, apply (20) for $\Delta\mathbf{z}_{[k]} = \Delta\mathbf{x}_{[k]}$ for all $k$ and sum them up gives

$$\frac{\sigma'}{2\tau} \sum_{k=1}^{K} \left\|\mathbf{A}(\Delta\mathbf{x} - \Delta\mathbf{x}^\star)_{[k]}\right\|_2^2 \leq \mathcal{H}_A(\Delta\mathbf{x}; \{\mathbf{v}_k\}) - \mathcal{H}_A(\Delta\mathbf{x}^\star; \{\mathbf{v}_k\}) \tag{22}$$

By Assumption 1 the previous inequality becomes

$$\frac{\sigma'}{2\tau} \sum_{k=1}^{K} \left\|\mathbf{A}(\Delta\mathbf{x} - \Delta\mathbf{x}^\star)_{[k]}\right\|_2^2 \leq \Theta(\mathcal{H}_A(\mathbf{0}; \{\mathbf{v}_k\}) - \mathcal{H}_A(\Delta\mathbf{x}^\star; \{\mathbf{v}_k\})) \tag{23}$$

The following inequality is straightforward

$$\frac{1}{2} \sum_{k=1}^{K} \left\|\mathbf{A}\Delta\mathbf{x}_{[k]}\right\|_2^2 \leq \sum_{k=1}^{K} \left\|\mathbf{A}\Delta\mathbf{x}_{[k]}^\star\right\|_2^2 + \sum_{k=1}^{K} \left\|\mathbf{A}(\Delta\mathbf{x} - \Delta\mathbf{x}^\star)_{[k]}\right\|_2^2 \tag{24}$$

Multiply (24) with $\sigma'/(2\tau)$ and use (21) and (23)

$$\frac{\sigma'}{4\tau} \sum_{k=1}^{K} \left\|\mathbf{A}\Delta\mathbf{x}_{[k]}\right\|_2^2 \leq (1 + \Theta)(\mathcal{H}_A(\mathbf{0}; \{\mathbf{v}_k\}) - \mathcal{H}_A(\Delta\mathbf{x}^\star; \{\mathbf{v}_k\})) \tag{25}$$

$\square$

## C.2 Main lemmas

We first present two main lemmas for the per-iteration improvement.

**Lemma 6.** *Let $g_i$ be strongly convex with convexity parameter $\mu_g \geq 0$ with respect to the norm $\|\cdot\|$, $\forall\, i \in [n]$. Then for all iterations $t$ of outer loop, and any $s \in [0,1]$, it holds that*

$$\mathbb{E}\left[\mathcal{H}_A(\mathbf{x}^{(t)};\mathbf{v}_k^{(t)}) - \mathcal{H}_A(\mathbf{x}^{(t+1)};\mathbf{v}_k^{(t+1)}) - \alpha\frac{\gamma\sigma_1'}{2\tau}\sum_{k=1}^{K}\left\|\mathbf{A}\Delta\mathbf{x}_{[k]}^{(t)}\right\|_2^2\right] \tag{26}$$

$$\geq \eta\left(sG_{\mathcal{H}}(\mathbf{x}^{(t)};\{\mathbf{v}_k'^{(t)}\}_{k=1}^{K}) - \frac{s^2\bar{\sigma}'}{2\tau}R^{(t)}\right) - \frac{9\beta\eta}{2\tau\sigma'}\sum_{k=1}^{K}\left\|\mathbf{v}_k^{(t)} - \mathbf{A}\mathbf{x}^{(t)}\right\|_2^2$$

*where $\alpha \in [0,1]$ is a constant and $\eta := \gamma(1-\Theta)(1-\alpha)$ and $\sigma_1' := \frac{(1-\Theta)}{2(1+\Theta)}\sigma'$ and $\bar{\sigma}' := (1+\beta)\sigma'$ and $\mathbf{v}_k' := \sum_{l=1}^{K}\mathcal{W}_{kl}\mathbf{v}_l$.*

$$R^{(t)} := -\frac{\tau\mu_g(1-s)}{\bar{\sigma}'s}\left\|\mathbf{u}^{(t)} - \mathbf{x}^{(t)}\right\|^2 + \sum_{k=1}^{K}\left\|\mathbf{A}(\mathbf{u}^{(t)} - \mathbf{x}^{(t)})_{[k]}\right\|^2 \tag{27}$$

*for $\mathbf{u}^{(t)} \in \mathbb{R}^n$ with $\bar{\mathbf{g}}'^{(t)} := \frac{1}{K}\sum_{k=1}^{K}\nabla f(\mathbf{v}_k'^{(t)})$*

$$u_i^{(t)} \in \partial g_i^*(-\mathbf{A}_i^\top\bar{\mathbf{g}}'^{(t)}) \qquad k \text{ s.t. } i \in \mathcal{P}_k \tag{28}$$

*Proof of Lemma 6.* For simplicity, we write $\mathcal{H}_A^{(t)}$ instead of $\mathcal{H}_A(\mathbf{x}^{(t)};\{\mathbf{v}_k^{(t)}\}_{k=1}^{K})$ and $\mathbf{v}_k' := \sum_{i=1}^{K}\mathcal{W}_{ik}\mathbf{v}_i$.

$$\mathbb{E}[\mathcal{H}_A^{(t)} - \mathcal{H}_A^{(t+1)}]$$

$$= \frac{1}{K}\sum_{k=1}^{K}f(\mathbf{v}_k) - \frac{1}{K}\sum_{k=1}^{K}f\left(\mathbf{v}_k' + \gamma K\Delta\mathbf{v}_{[k]}\right) + g(\mathbf{x}) - g(\mathbf{x} + \gamma\Delta\mathbf{x})$$

$$= \underbrace{\frac{1}{K}\sum_{k=1}^{K}f(\mathbf{v}_k) - \frac{1}{K}\sum_{k=1}^{K}f\left(\mathbf{v}_k'\right)}_{D_1}$$

$$+ \underbrace{\sum_{k=1}^{K}\left\{\frac{1}{K}f\left(\mathbf{v}_k'\right) + g(\mathbf{x}_{[k]})\right\} - \sum_{k=1}^{K}\left\{\frac{1}{K}f\left(\mathbf{v}_k' + \gamma K\Delta\mathbf{v}_{[k]}\right) + g(\mathbf{x}_{[k]} + \gamma\Delta\mathbf{x}_{[k]})\right\}}_{D_2}$$

By the convexity of $f$, $D_1 \geq 0$. Using the convexity of $f$ and $g$ in $D_2$ we have

$$\frac{1}{\gamma}D_2 \geq \sum_{k=1}^{K}\left\{\frac{1}{K}f\left(\mathbf{v}_k'\right) + g(\mathbf{x}_{[k]})\right\} - \sum_{k=1}^{K}\left\{\frac{1}{K}f\left(\mathbf{v}_k' + K\Delta\mathbf{v}_{[k]}\right) + g(\mathbf{x}_{[k]} + \Delta\mathbf{x}_{[k]})\right\}$$

$$\geq \mathbb{E}\left[\sum_{k=1}^{K}\mathscr{G}_k^{\sigma'}\left(\mathbf{0};\mathbf{v}_k',\mathbf{x}_{[k]}\right) - \sum_{k=1}^{K}\mathscr{G}_k^{\sigma'}\left(\Delta\mathbf{x}_{[k]};\mathbf{v}_k',\mathbf{x}_{[k]}\right)\right]$$

Use the Assumption 1 we have

$$D_2 \geq \gamma\mathbb{E}\left[\sum_{k=1}^{K}\mathscr{G}_k^{\sigma'}\left(\mathbf{0};\mathbf{v}_k',\mathbf{x}_{[k]}\right) - \sum_{k=1}^{K}\mathscr{G}_k^{\sigma'}\left(\Delta\mathbf{x}_{[k]};\mathbf{v}_k',\mathbf{x}_{[k]}\right)\right]$$

$$\geq \gamma(1-\Theta)\underbrace{\sum_{k=1}^{K}\left\{\mathscr{G}_k^{\sigma_2'}\left(\mathbf{0};\mathbf{v}_k',\mathbf{x}_{[k]}\right) - \mathscr{G}_k^{\sigma_2'}\left(\Delta\mathbf{x}_{[k]}^\star;\mathbf{v}_k',\mathbf{x}_{[k]}\right)\right\}}_{C}$$

Let $\alpha \in [0,1]$ and apply Lemma 5, the previous inequality becomes

$$D_2 \geq \gamma(1-\Theta)(1-\alpha)C + \alpha\frac{\gamma\sigma_1'}{2\tau}\sum_{k=1}^{K}\left\|\mathbf{A}\Delta\mathbf{x}_{[k]}\right\|_2^2 \tag{29}$$

where $\sigma'_1 := \frac{(1-\Theta)}{2(1+\Theta)}\sigma'$. From the definition of $u_i$ we know

$$g_i(u_i) = u_i(-\mathbf{A}_i^\top \bar{\mathbf{g}}') - g_i^*(-\mathbf{A}_i^\top \bar{\mathbf{g}}') \tag{30}$$

Replacing $\Delta\mathbf{x}_i = s(u_i - x_i)$ in $C$ gives

$$C \geq \sum_{k=1}^{K} \left\{ \sum_{i \in \mathcal{P}_k} (g_i(x_i) - g_i(x_i + \Delta x_i)) - \langle \mathbf{g}'_k, \mathbf{A}\Delta\mathbf{x}_{[k]} \rangle - \frac{\sigma'}{2\tau} \left\| \mathbf{A}\Delta\mathbf{x}_{[k]} \right\|^2 \right\}$$

$$\geq \sum_{k=1}^{K} \left\{ \sum_{i \in \mathcal{P}_k} (sg_i(x_i) - sg_i(u_i) + \frac{\mu_g}{2\tau}(1-s)s(u_i - x_i)^2) - \langle \mathbf{g}'_k, \mathbf{A}\Delta\mathbf{x}_{[k]} \rangle - \frac{\sigma'}{2\tau} \left\| \mathbf{A}\Delta\mathbf{x}_{[k]} \right\|^2 \right\}$$

$$\overset{(30)}{=} \sum_{k=1}^{K} \left( \sum_{i \in \mathcal{P}_k} \left( sg_i(x_i) + sg_i^*(-\mathbf{A}_i^\top \bar{\mathbf{g}}') \right) + s\langle \mathbf{v}_k/K, \mathbf{g}'_k \rangle \right) - s \sum_{k=1}^{K} (\langle \mathbf{v}_k/K, \mathbf{g}'_k \rangle - \langle \mathbf{A}\mathbf{u}_{[k]}, \bar{\mathbf{g}}' \rangle)$$

$$+ \sum_{k=1}^{K} \sum_{i \in \mathcal{P}_k} \left\{ \frac{\mu_g}{2\tau}(1-s)s(u_i - x_i)^2 \right\} - \sum_{k=1}^{K} \langle \mathbf{g}'_k, \mathbf{A}\Delta\mathbf{x}_{[k]} \rangle - \sum_{k=1}^{K} \frac{\sigma'}{2\tau} \left\| \mathbf{A}\Delta\mathbf{x}_{[k]} \right\|^2$$

$$= \sum_{k=1}^{K} \left\{ \sum_{i \in \mathcal{P}_k} \left( sg_i(x_i) + sg_i^*(-\mathbf{A}_i^\top \bar{\mathbf{g}}') \right) + s\langle \mathbf{v}_k/K, \mathbf{g}'_k \rangle \right\} + \frac{\mu_g}{2}(1-s)s \left\| \mathbf{u} - \mathbf{x} \right\|^2$$

$$- \sum_{k=1}^{K} \frac{s^2 \sigma'}{2\tau} \left\| \mathbf{A}(\mathbf{u} - \mathbf{x})_{[k]} \right\|^2 - s \sum_{k=1}^{K} (\langle \mathbf{v}_k/K, \mathbf{g}'_k \rangle - \langle \mathbf{A}\mathbf{u}_{[k]}, \bar{\mathbf{g}}' \rangle + \langle \mathbf{g}'_k, \mathbf{A}(\mathbf{u} - \mathbf{x})_{[k]} \rangle)$$

$$= sG_{\mathcal{H}}(\mathbf{x}; \{\mathbf{v}'_k\}_{k=1}^{K}) + \frac{\mu_g}{2}(1-s)s \left\| \mathbf{u} - \mathbf{x} \right\|^2$$

$$- \frac{s^2 \sigma'}{2\tau} \sum_{k=1}^{K} \left\| \mathbf{A}(\mathbf{u} - \mathbf{x})_{[k]} \right\|^2 - s \sum_{k=1}^{K} (\langle \mathbf{v}'_k/K, \mathbf{g}'_k \rangle - \langle \mathbf{A}\mathbf{u}_{[k]}, \bar{\mathbf{g}}' \rangle + \langle \mathbf{g}'_k, \mathbf{A}(\mathbf{u} - \mathbf{x})_{[k]} \rangle)$$

We can bound the last term of the previous equation as $D_3$

$$\frac{1}{s}D_3 = \sum_{k=1}^{K} (\langle \mathbf{v}'_k/K, \mathbf{g}'_k \rangle - \langle \mathbf{A}\mathbf{u}_{[k]}, \bar{\mathbf{g}}' \rangle + \langle \mathbf{g}'_k, \mathbf{A}(\mathbf{u} - \mathbf{x})_{[k]} \rangle)$$

$$= \sum_{k=1}^{K} (\langle \mathbf{g}'_k, \mathbf{v}'_k/K \rangle - \langle \mathbf{g}'_k, \mathbf{A}\mathbf{u}/K \rangle + \langle \mathbf{g}'_k, \mathbf{A}(\mathbf{u} - \mathbf{x})_{[k]} \rangle)$$

$$= \frac{1}{K} \sum_{k=1}^{K} \langle \mathbf{g}'_k, \mathbf{v}'_k - \mathbf{A}\mathbf{x} \rangle - \sum_{k=1}^{K} \langle \bar{\mathbf{g}}', \mathbf{A}(\mathbf{u} - \mathbf{x})_{[k]} \rangle + \sum_{k=1}^{K} \langle \mathbf{g}'_k, \mathbf{A}(\mathbf{u} - \mathbf{x})_{[k]} \rangle$$

$$= \frac{1}{K} \sum_{k=1}^{K} \langle \mathbf{g}'_k - \bar{\mathbf{g}}', \mathbf{v}'_k - \mathbf{A}\mathbf{x} \rangle + \sum_{k=1}^{K} \langle \mathbf{g}'_k - \bar{\mathbf{g}}', \mathbf{A}(\mathbf{u} - \mathbf{x})_{[k]} \rangle$$

Bound the gradient terms with consensus violation. First bound $\sum_{k=1}^{K} \|\mathbf{g}'_k - \bar{\mathbf{g}}'\|_2^2$, define $\mathbf{g}_c := \nabla f(\mathbf{A}\mathbf{x})$

$$\sum_{k=1}^{K} \|\mathbf{g}'_k - \bar{\mathbf{g}}'\|_2^2 \leq 2 \sum_{k=1}^{K} \left( \|\mathbf{g}'_k - \mathbf{g}_c\|_2^2 + \|\mathbf{g}_c - \bar{\mathbf{g}}'\|_2^2 \right) \leq 2 \sum_{k=1}^{K} \|\mathbf{g}'_k - \mathbf{g}_c\|_2^2 + 2\frac{1}{K} \sum_{k=1}^{K} \|\mathbf{g}_c - \mathbf{g}'_k\|_2^2$$

Apply the $1/\tau$-smoothness of $f$ we have

$$\sum_{k=1}^{K} \|\mathbf{g}'_k - \bar{\mathbf{g}}'\|_2^2 \leq \frac{4}{\tau^2} \sum_{k=1}^{K} \|\mathbf{v}'_k - \mathbf{A}\mathbf{x}\|_2^2 \leq \frac{4\beta^2}{\tau^2} \sum_{k=1}^{K} \|\mathbf{v}_k - \mathbf{A}\mathbf{x}\|_2^2 \tag{31}$$

Bound the first term in $D_3$

$$s\frac{1}{K} \sum_{k=1}^{K} \langle \mathbf{g}'_k - \bar{\mathbf{g}}', \mathbf{v}'_k - \mathbf{A}\mathbf{x} \rangle \leq \frac{s}{2K} \sum_{k=1}^{K} \left( \tau \|\mathbf{g}'_k - \bar{\mathbf{g}}'\|_2^2 + \frac{1}{\tau} \|\mathbf{v}'_k - \mathbf{A}\mathbf{x}\|_2^2 \right)$$

$$\overset{(31)}{\leq} \frac{5\beta^2 s}{2\tau K} \sum_{k=1}^{K} \|\mathbf{v}_k - \mathbf{Ax}\|_2^2$$

Bound the second term in $D_3$

$$s\sum_{k=1}^{K}\langle \mathbf{g}_k' - \bar{\mathbf{g}}', \mathbf{A}(\mathbf{u}-\mathbf{x})_{[k]}\rangle \leq \frac{\tau}{2\sigma'\beta}\sum_{k=1}^{K}\|\mathbf{g}_k' - \bar{\mathbf{g}}'\|_2^2 + \frac{s^2\sigma'\beta}{2\tau}\sum_{k=1}^{K}\left\|\mathbf{A}(\mathbf{u}-\mathbf{x})_{[k]}\right\|_2^2$$

$$\overset{(31)}{\leq} \frac{4\beta}{2\tau\sigma'}\sum_{k=1}^{K}\|\mathbf{v}_k - \mathbf{Ax}\|_2^2 + \frac{s^2\sigma'\beta}{2\tau}\sum_{k=1}^{K}\left\|\mathbf{A}(\mathbf{u}-\mathbf{x})_{[k]}\right\|_2^2$$

Then

$$C \geq sG_{\mathcal{H}}(\mathbf{x}, \{\mathbf{v}_k'\}_{k=1}^{K}) + \frac{\mu_g}{2}(1-s)s\,\|\mathbf{u}-\mathbf{x}\|^2 - \frac{s^2(\sigma'+\beta\sigma')}{2\tau}\sum_{k=1}^{K}\left\|\mathbf{A}(\mathbf{u}-\mathbf{x})_{[k]}\right\|^2$$

$$- \frac{9\beta}{2\tau\sigma'}\sum_{k=1}^{K}\|\mathbf{v}_k - \mathbf{Ax}\|_2^2$$

Then let $\bar{\sigma}' := (1+\beta)\sigma'$ and $\eta := \gamma(1-\Theta)(1-\alpha)$ we have

$$\mathbb{E}[\mathcal{H}_A^{(t)} - \mathcal{H}_A^{(t+1)} - \alpha\frac{\gamma\sigma_1'}{2\tau}\sum_{k=1}^{K}\left\|\mathbf{A}\Delta\mathbf{x}_{[k]}^{(t)}\right\|_2^2]$$

$$\geq \eta\left(sG_{\mathcal{H}}(\mathbf{x}^{(t)}; \{\mathbf{v}_k'^{(t)}\}_{k=1}^{K}) - \frac{s^2\bar{\sigma}'}{2\tau}R^{(t)}\right) - \frac{9\eta\beta}{2\tau\sigma'}\sum_{k=1}^{K}\left\|\mathbf{v}_k^{(t)} - \mathbf{Ax}^{(t)}\right\|_2^2$$

$\square$

The following lemma correlates the consensus violation with the size of updates

**Lemma 7.** *Let $c > 0$ be any constant value. Define $\delta^{(0)} := \mathbf{0}$ and*

$$\delta^{(t+1)} := \beta\delta^{(t)} + cc_1\sum_{k=1}^{K}\left\|\Delta\mathbf{v}_k^{(t)}\right\|_2^2 \tag{32}$$

*Then the consensus violation has an upper bound.*

$$\sum_{k=1}^{K}\left\|\mathbf{v}_k^{(t)} - \mathbf{v}^{(t)}\right\|_2^2 \leq e_1\delta^{(t)} \tag{33}$$

*where $e_1 := (1-\beta)/c$.*

*Proof.* Let

$$a_t := \sum_{k=1}^{K}\left\|\Delta\mathbf{v}_k^{(t)}\right\|_2^2, \, b_t := \sum_{k=1}^{K}\left\|\mathbf{v}_k^{(t)} - \mathbf{v}^{(t)}\right\|_2^2 \tag{34}$$

We want to prove that

$$b_t \leq e_1\delta^{(t)} \tag{35}$$

First $t = 0$, $b_0 = \delta^{(0)} = 0$. If the claim holds for time $t - 1$, then $b_{t-1} \leq e_1\delta^{(t-1)}$. At time $t$, we have

$$b_t \overset{(14)}{\leq} \beta b_{t-1} + (1-\beta)c_1 a_{t-1} \tag{36}$$

$$\leq \beta\frac{1-\beta}{c}\delta^{(t-1)} + (1-\beta)c_1 a_{t-1} \tag{37}$$

$$\leq \frac{1-\beta}{c}(\beta\delta^{(t-1)} + cc_1 a_{t-1}) \tag{38}$$

$$\overset{(32)}{\leq} e_1\delta^{(t)} \tag{39}$$

Thus we proved the lemma. $\square$

**Lemma 8.** *Let $g_i$ be strongly convex with convexity parameter $\mu_g \geq 0$ with respect to the norm $\|\cdot\|, \forall\, i \in [n]$. Then for all iterations $t$ of outer loop, and $s \in [0,1]$, it holds that*

$$\mathbb{E}[\mathcal{H}_A(\mathbf{x}^{(t)}; \{\mathbf{v}_k^{(t)}\}_{k=1}^K) - \mathcal{H}_A(\mathbf{x}^{(t+1)}; \{\mathbf{v}_k^{(t+1)}\}_{k=1}^K) + \frac{1+\beta}{2}\delta^{(t)} - \delta^{(t+1)}]$$

$$\geq \eta\left(sG_{\mathcal{H}}(\mathbf{x}^{(t)}; \{\textstyle\sum_{i=1}^K \mathcal{W}_{ki}\mathbf{v}_i^{(t)}\}_{k=1}^K) - \frac{s^2\bar{\sigma}'}{2\tau}R^{(t)}\right) \tag{40}$$

*where $\alpha := (1 + \frac{(1-\beta)^2}{36(1+\Theta)\beta})^{-1} \in [0,1]$, $\eta := \gamma(1-\Theta)(1-\alpha)$, $\bar{\sigma}' := (1+\beta)\sigma'$ and*

$$R^{(t)} := -\frac{\tau\mu_g(1-s)}{\bar{\sigma}'s}\left\|\mathbf{u}^{(t)} - \mathbf{x}^{(t)}\right\|^2 + \textstyle\sum_{k=1}^K \left\|\mathbf{A}(\mathbf{u}^{(t)} - \mathbf{x}^{(t)})_{[k]}\right\|^2 \tag{41}$$

*for $\mathbf{u}^{(t)} \in \mathbb{R}^n$ with $\bar{\mathbf{g}}'^{(t)} := \sum_{k=1}^K \nabla f(\sum_{i=1}^K \mathcal{W}_{ik}\mathbf{v}_i^{(t)})$*

$$u_i^{(t)} \in \partial g_i^*(-\mathbf{A}_i^\top \bar{\mathbf{g}}'^{(t)}) \qquad k \text{ s.t. } i \in \mathcal{P}_k. \tag{42}$$

*where $\delta^{(t)}$ is defined in Lemma 7.*

*Proof.* In this proof we use $\mathbf{v}_k' := \sum_i \mathcal{W}_{ki}\mathbf{v}_i$. From Lemma 6 we know that

$$\mathbb{E}\left[\mathcal{H}_A^{(t)} - \mathcal{H}_A^{(t+1)} - \alpha\frac{\gamma\sigma_1'}{2\tau}\sum_{k=1}^K\left\|\mathbf{A}\Delta\mathbf{x}_{[k]}^{(t)}\right\|_2^2 + \frac{9\eta\beta}{2\tau\sigma'}\sum_{k=1}^K\left\|\mathbf{v}_k^{(t)} - \mathbf{A}\mathbf{x}^{(t)}\right\|_2^2\right]$$

$$\geq \eta\left(sG_{\mathcal{H}}(\mathbf{x}^{(t)}; \{\mathbf{v}_k'^{(t)}\}_{k=1}^K) - \frac{s^2\bar{\sigma}'}{2\tau}R^{(t)}\right)$$

Use the following notations to simplify the calculation

$$a_t := \sum_{k=1}^K\left\|\Delta\mathbf{v}_k^{(t)}\right\|_2^2, b_t := \sum_{k=1}^K\left\|\mathbf{v}_k^{(t)} - \mathbf{v}^{(t)}\right\|_2^2, f_1 := \alpha\frac{\gamma\sigma_1'}{2\tau}, f_2 := \frac{9\eta\beta}{2\tau\sigma'} \tag{43}$$

From Lemma 7 we know that

$$f_2 b_t - f_1 a_t \leq f_2 e_1\delta^{(t)} - f_1(\delta^{(t+1)} - \beta\delta^{(t)})/(cc_1) = (f_2 e_1 + \frac{f_1\beta}{cc_1})\delta^{(t)} - \frac{f_1}{cc_1}\delta^{(t+1)} \tag{44}$$

Fix constant $c$ such that $\frac{f_1}{cc_1} = 1$ in (44)

$$c = \frac{f_1}{c_1} = \alpha\frac{(1-\beta)^2\sigma_1'}{2\tau\gamma K^2} \tag{45}$$

Fix $(f_2 e_1 + \frac{f_1\beta}{cc_1}) = \frac{1+\beta}{2} < 1$ in (44), to determine $\alpha \in [0,1]$. First consider $f_2 e_1$

$$f_2 e_1 = \frac{9\gamma(1-\Theta)(1-\alpha)\beta}{2\tau\sigma'}\frac{1-\beta}{c} \overset{(45)}{=} \frac{1-\alpha}{\alpha}\frac{9(1-\Theta)\beta}{1-\beta}\frac{\gamma K}{\sigma_1'} \tag{46}$$

Then we have

$$f_2 e_1 + \frac{f_1\beta}{cc_1} = \frac{1-\alpha}{\alpha}\frac{9(1-\Theta)\beta}{1-\beta}\frac{\gamma K}{\sigma_1'} + \beta = \frac{1+\beta}{2} < 1. \tag{47}$$

Thus we can fix $\alpha \in [0,1]$ to be

$$\alpha := \left(1 + \frac{(1-\beta)^2}{36(1+\Theta)\beta}\right)^{-1} \tag{48}$$

So when have these information.

$$f_2 b_t - f_1 a_t \leq \frac{1+\beta}{2}\delta^{(t)} - \delta^{(t+1)} \tag{49}$$

Finally, using all of the previous equations we know

$$\mathbb{E}\left[\mathcal{H}_A^{(t)} - \mathcal{H}_A^{(t+1)} + \frac{1+\beta}{2}\delta^{(t)} - \delta^{(t+1)}\right] \geq \eta\left(sG_{\mathcal{H}}(\mathbf{x}^{(t)}; \{\mathbf{v}_k'^{(t)}\}_{k=1}^K) - \frac{s^2\bar{\sigma}'}{2\tau}R^{(t)}\right) \tag{50}$$

$$\square$$

## C.3  Main theorems

Here we present the proofs of Theorem 1 and Theorem 2.

**Lemma 9.** *If $g_i^*$ are L-Lipschitz continuous for all $i \in [n]$, then*

$$\forall\, t : R^{(t)} \leq 4L^2 \sum_{k=1}^{K} \sigma_k n_k = 4L^2 \sigma, \tag{51}$$

*where $\sigma_k := \max_{\mathbf{x}_{[k]} \in \mathbb{R}^n} \left\| \mathbf{A}_{[k]} \mathbf{x}_{[k]} \right\|^2 / \left\| \mathbf{x}_{[k]} \right\|^2$.*

*Proof.* For general convex functions, the strong convexity parameter is $\mu_g = 0$, and hence the definition (41) of the complexity constant $R^{(t)}$ becomes

$$R^{(t)} = \sum_{k=1}^{K} \left\| \mathbf{A}(\mathbf{u}^{(t)} - \mathbf{x}^{(t)})_{[k]} \right\|^2 = \sum_{k=1}^{K} \sigma_k \left\| (\mathbf{u}^{(t)} - \mathbf{x}^{(t)})_{[k]} \right\|^2 \leq \sum_{k=1}^{K} \sigma_k |\mathcal{P}_k| 4L^2 = 4L^2 \sigma$$

Here the last inequality follows from $L$-Lipschitz property of $g^*$. $\qquad\square$

**Theorem 1** (Strongly Convex $g_i$). *Consider Algorithm 1 with $\gamma := 1$ and let $\Theta$ be the quality of the local solver in Assumption 1. Let $g_i$ be $\mu_g$-strongly convex for all $i \in [n]$ and let $f$ be $1/\tau$-smooth. Let $\bar{\sigma}' := (1 + \beta)\sigma'$, $\alpha := (1 + \frac{(1-\beta)^2}{36(1+\Theta)\beta})^{-1}$ and $\eta := \gamma(1 - \Theta)(1 - \alpha)$*

$$s_0 = \frac{\tau \mu_g}{\tau \mu_g + \sigma_{max}\bar{\sigma}'} \in [0, 1]. \tag{8}$$

*Then after $T$ iterations of Algorithm 1 with[5]*

$$T \geq \frac{1 + \eta s_0}{\eta s_0} \log \frac{\varepsilon_{\mathcal{H}}^{(0)}}{\varepsilon_{\mathcal{H}}},$$

*it holds that $\mathbb{E}\big[\mathcal{H}_A(\mathbf{x}^{(T)}, \{\mathbf{v}_k^{(T)}\}_{k=1}^K) - \mathcal{H}_A(\mathbf{x}^\star, \{\mathbf{v}_k^\star\}_{k=1}^K)\big] \leq \varepsilon_{\mathcal{H}}$. Furthermore, after $T$ iterations with*

$$T \geq \frac{1 + \eta s_0}{\eta s_0} \log \left( \frac{1}{\eta s_0} \frac{\varepsilon_{\mathcal{H}}^{(0)}}{\varepsilon_{G_{\mathcal{H}}}}, \right)$$

*we have the expected duality gap $\mathbb{E}[G_{\mathcal{H}}(\mathbf{x}^{(T)}, \{\sum_{k=1}^K \mathcal{W}_{kl}\mathbf{v}_l^{(T)}\}_{k=1}^K)] \leq \varepsilon_{G_{\mathcal{H}}}$.*

*Proof.* If $g_i(\cdot)$ are $\mu_g$-strongly convex, one can use the definition of $\sigma_k$ and $\sigma_{\max}$ to find

$$\begin{aligned}
R^{(t)} &\leq -\frac{\tau \mu_g(1 - s)}{\bar{\sigma}' s} \left\| \mathbf{u}^{(t)} - \mathbf{x}^{(t)} \right\|^2 + \sum_{k=1}^{K} \left\| \mathbf{A}(\mathbf{u}^{(t)} - \mathbf{x}^{(t)})_{[k]} \right\|^2 \\
&\leq \left( -\frac{\tau \mu_g(1 - s)}{\bar{\sigma}' s} + \sigma_{\max} \right) \left\| \mathbf{u}^{(t)} - \mathbf{x}^{(t)} \right\|^2.
\end{aligned} \tag{52}$$

If we set

$$s_0 = \frac{\tau \mu_g}{\tau \mu_g + \sigma_{\max}\bar{\sigma}'} \tag{53}$$

then $R^{(t)} \leq 0$. The duality gap has a lower bound duality gap

$$G_{\mathcal{H}}(\mathbf{x}^{(t)}, \{\mathbf{v}_k'^{(t)}\}_{k=1}^K) \geq \mathcal{H}_A(\mathbf{x}^{(t)}, \{\mathbf{v}_k'^{(t)}\}_{k=1}^K) - \mathcal{H}_A^* \geq \mathcal{H}_A(\mathbf{x}^{(t+1)}, \{\mathbf{v}_k^{(t+1)}\}_{k=1}^K) - \mathcal{H}_A^* \tag{54}$$

and use Lemma 8, we have

$$\mathbb{E}[\mathcal{H}_A^{(t)} - \mathcal{H}_A^{(t+1)} + \frac{1 + \beta}{2}\delta^{(t)} - \delta^{(t+1)}] \geq \eta s_0 G_{\mathcal{H}} \geq \eta s_0(\mathcal{H}_A^{(t+1)} - \mathcal{H}_A^\star) \tag{55}$$

Then

$$\mathbb{E}[\mathcal{H}_A^{(t)} - \mathcal{H}_A^\star + \frac{1 + \beta}{2}\delta^{(t)}] \geq (1 + \eta s_0)\mathbb{E}[\mathcal{H}_A^{(t+1)} - \mathcal{H}_A^\star + \delta^{(t+1)}] \tag{56}$$

Therefore if we denote $\varepsilon_{\mathcal{H}}^{(t)} := \mathcal{H}_A^{(t)} - \mathcal{H}_A^\star + \delta^{(t)}$ we have recursively that

$$\mathbb{E}[\varepsilon_{\mathcal{H}}^{(t)}] \leq \left(1 - \frac{\eta s_0}{1 + \eta s_0}\right)^t \varepsilon_{\mathcal{H}}^{(0)} \leq \exp\left(-\frac{\eta s_0}{1 + \eta s_0} t\right) \varepsilon_{\mathcal{H}}^{(0)}$$

The right hand side will be smaller than some $\varepsilon_{\mathcal{H}}$ if

$$T \geq \frac{1 + \eta s_0}{\eta s_0} \log \frac{\varepsilon_{\mathcal{H}}^{(0)}}{\varepsilon_{\mathcal{H}}}$$

Moreover, to bound the duality gap $G_{\mathcal{H}}^{(t)}$, we have

$$\eta s_0 G_{\mathcal{H}}^{(t)} \overset{(55)}{\leq} \mathbb{E}[\mathcal{H}_A^{(t)} - \mathcal{H}_A^{(t+1)} + \frac{1+\beta}{2}\delta^{(t)} - \delta^{(t+1)}] \leq \mathbb{E}[\mathcal{H}_A^{(t)} - \mathcal{H}_A^* + \delta^{(t)}]$$

Hence if $\varepsilon_{\mathcal{H}} \leq \eta s_0 \varepsilon_{G_{\mathcal{H}}}$ then $G_{\mathcal{H}}^{(t)} \leq \varepsilon_{G_{\mathcal{H}}}$. Therefore after

$$T \geq \frac{1 + \eta s_0}{\eta s_0} \log\left(\frac{1}{\eta s_0} \frac{\varepsilon_{\mathcal{H}}^{(0)}}{\varepsilon_{G_{\mathcal{H}}}}\right)$$

iterations we have obtained a duality gap less then $\varepsilon_{G_{\mathcal{H}}}$.

$\square$

**Theorem 2** (Non-strongly Convex Case). *Consider Algorithm 1, using a local solver of quality $\Theta$. Let $g_i(\cdot)$ have $L$-bounded support, and let $f$ be $(1/\tau)$-smooth. Let $\varepsilon_{G_{\mathcal{H}}} > 0$ be the desired duality gap. Then after $T$ iterations where*

$$T \geq T_0 + \max\left\{\left\lceil\frac{1}{\eta}\right\rceil, \frac{4L^2\sigma\bar{\sigma}'}{\tau\varepsilon_{G_{\mathcal{H}}}\eta}\right\}, \qquad T_0 \geq t_0 + \left[\frac{2}{\eta}\left(\frac{8L^2\sigma\bar{\sigma}'}{\tau\varepsilon_{G_{\mathcal{H}}}} - 1\right)\right]_+$$

$$t_0 \geq \max\left\{0, \left\lceil\frac{1+\eta}{\eta} \log \frac{2\tau(\mathcal{H}_A(\mathbf{x}^{(0)}, \{\mathbf{v}_i^{(0)}\}) - \mathcal{H}_A(\mathbf{x}^\star, \{\mathbf{v}^\star\}))}{4L^2\sigma\bar{\sigma}'}\right\rceil\right\}$$

*and $\bar{\sigma}' := (1+\beta)\sigma'$, $\alpha := (1 + \frac{(1-\beta)^2}{36(1+\Theta)\beta})^{-1}$ and $\eta := \gamma(1-\Theta)(1-\alpha)$. We have that the expected duality gap satisfies*

$$\mathbb{E}\left[G_{\mathcal{H}}(\bar{\mathbf{x}}, \{\bar{\mathbf{v}}_k\}_{k=1}^K, \{\bar{\mathbf{w}}_k\}_{k=1}^K)\right] \leq \varepsilon_{G_{\mathcal{H}}}$$

*at the averaged iterate $\bar{\mathbf{x}} := \frac{1}{T-T_0}\sum_{t=T_0+1}^{T-1} \mathbf{x}^{(t)}$, and $\mathbf{v}_k' := \sum_{l=1}^K \mathcal{W}_{kl}\mathbf{v}_l$ and $\bar{\mathbf{v}}_k := \frac{1}{T-T_0}\sum_{t=T_0+1}^{T-1}(\mathbf{v}_k')^{(t)}$ and $\bar{\mathbf{w}}_k := \frac{1}{T-T_0}\sum_{t=T_0+1}^{T-1} \nabla f((\mathbf{v}_k')^{(t)})$.*

*Proof.* We write $\mathcal{H}_A^{(t)}$ instead of $\mathcal{H}_A(\mathbf{x}^{(t)}; \{\mathbf{v}_k^{(t)}\}_{k=1}^K)$ and $\mathcal{H}_A^\star$ instead of $\mathcal{H}_A(\mathbf{x}^\star; \{\mathbf{v}_k^\star\}_{k=1}^K)$. We begin by estimating the expected change of feasibility for $\mathcal{H}_A$. We can bound this above by using Lemma 8 and the fact that $F_B(\cdot)$ is always a lower bound for $-F_A(\cdot)$ and then applying (51) to find

$$(1 + \eta s)\mathbb{E}[\mathcal{H}_A^{(t+1)} - \mathcal{H}_A^\star + \delta^{(t+1)}] \leq (\mathcal{H}_A^{(t)} - \mathcal{H}_A^\star + \delta^{(t)}) + \eta \frac{\bar{\sigma}' s^2}{2\tau} 4L^2\sigma \tag{57}$$

Use (57) recursively we have

$$\mathbb{E}[\mathcal{H}_A^{(t)} - \mathcal{H}_A^\star + \delta^{(t)}] \leq (1 + \eta s)^{-t}(\mathcal{H}_A^{(0)} - \mathcal{H}_A^\star + \delta^{(0)}) + s\frac{4L^2\bar{\sigma}'\sigma}{2\tau} \tag{58}$$

We know that $\delta^{(0)} = 0$. Choose $s = 1$ and

$$t = t_0 := \max\left\{0, \left\lceil\frac{1+\eta}{\eta} \log \frac{2\tau(\mathcal{H}_A^{(0)} - \mathcal{H}_A^\star)}{4L^2\sigma\bar{\sigma}'}\right\rceil\right\} \tag{59}$$

leads to

$$\mathbb{E}[\mathcal{H}_A^{(t)} - \mathcal{H}_A^\star + \delta^{(t)}] \leq \frac{4L^2\bar{\sigma}'\sigma}{\tau} \tag{60}$$

Next, we show inductively that

$$\forall\, t \geq t_0 : \mathbb{E}[\mathcal{H}_A^{(t)} - \mathcal{H}_A^\star + \delta^{(t)}] \leq \frac{4L^2\bar{\sigma}'\sigma}{\tau(1 + \frac{1}{2}\eta(t - t_0))}. \tag{61}$$

Clearly, (60) implies that (61) holds for $t = t_0$. Assuming that it holds for any $t \geq t_0$, we show that it must also hold for $t + 1$. Indeed, using

$$s = \frac{1}{1 + \frac{1}{2}\eta(t - t_0)} \in [0, 1], \tag{62}$$

we obtain

$$\mathbb{E}[\mathcal{H}_A^{(t+1)} - \mathcal{H}_A^\star + \delta^{(t+1)}] \leq \frac{4L^2\sigma\bar{\sigma}'}{\tau} \underbrace{\left( \frac{1 + \frac{1}{2}\eta(t - t_0) - \frac{1}{2}\gamma(1 - \Theta)}{(1 + \frac{1}{2}\eta(t - t_0))^2} \right)}_{D}$$

by applying the bounds (57) and (61), plugging in the definition of $s$ (62), and simplifying. We upper bound the term $D$ using the fact that geometric mean is less or equal to arithmetic mean:

$$D = \frac{1}{1 + \frac{1}{2}\eta(t + 1 - t_0)} \underbrace{\frac{(1 + \frac{1}{2}\eta(t + 1 - t_0))(1 + \frac{1}{2}\eta(t - 1 - t_0))}{(1 + \frac{1}{2}\eta(t - t_0))^2}}_{\leq 1}$$

$$\leq \frac{1}{1 + \frac{1}{2}\eta(t + 1 - t_0)}.$$

We can apply the results of Lemma 8 to get

$$\eta s G_{\mathcal{H}}(\mathbf{x}^{(t)}, \{\mathbf{v}_k^{(t)}\}_{k=1}^K) \leq \mathcal{H}_A^{(t)} - \mathcal{H}_A^{(t+1)} + \delta^{(t)} - \delta^{(t+1)}$$

Define the following iterate

$$\bar{\mathbf{x}} := \frac{1}{T - T_0} \sum_{t=T_0+1}^{T-1} \mathbf{x}^{(t)}, \quad \bar{\mathbf{v}}_k := \frac{1}{T - T_0} \sum_{t=T_0+1}^{T-1} \mathbf{v}_k^{(t)}, \quad \bar{\mathbf{w}}_k := \frac{1}{T - T_0} \sum_{t=T_0+1}^{T-1} \nabla f(\mathbf{v}_k^{(t)})$$

use Lemma 9 to obtain

$$\mathbb{E}[G_{\mathcal{H}}(\bar{\mathbf{x}}, \{\bar{\mathbf{v}}_k\}_{k=1}^K, \{\bar{\mathbf{w}}_k\}_{k=1}^K)] \leq \frac{1}{T - T_0} \sum_{t=T_0}^{T-1} \mathbb{E}[G_{\mathcal{H}}\left(\mathbf{x}^{(t)}, \{\mathbf{v}_k^{(t)}\}_{k=1}^K\right)]$$

$$\leq \frac{1}{\eta s} \frac{1}{T - T_0} \mathbb{E}[\mathcal{H}_A^{(T_0)} - \mathcal{H}_A^\star + \delta^{(T_0)}] + \frac{4L^2\sigma\bar{\sigma}'s}{2\tau}$$

If $T \geq \lceil \frac{1}{\eta} \rceil + T_0$ such that $T_0 \geq t_0$ we have

$$\mathbb{E}[G_{\mathcal{H}}(\bar{\mathbf{x}}, \{\bar{\mathbf{v}}_k\}_{k=1}^K, \{\bar{\mathbf{w}}_k\}_{k=1}^K)] \leq \frac{1}{\eta s} \frac{1}{T - T_0} \left( \frac{4L^2\bar{\sigma}'\sigma}{\tau(1 + \frac{1}{2}\eta(T_0 - t_0))} \right) + \frac{4L^2\sigma\bar{\sigma}'s}{2\tau}$$

$$= \frac{4L^2\sigma\bar{\sigma}'}{\tau} \left( \frac{1}{\eta s} \frac{1}{T - T_0} \frac{1}{(1 + \frac{1}{2}\eta(T_0 - t_0))} + \frac{s}{2} \right).$$

Choosing

$$s = \frac{1}{(T - T_0)\eta} \in [0, 1] \tag{63}$$

gives us

$$\mathbb{E}[G_{\mathcal{H}}(\bar{\mathbf{x}}, \{\bar{\mathbf{v}}_k\}_{k=1}^K, \{\bar{\mathbf{w}}_k\}_{k=1}^K)] \leq \frac{4L^2\sigma\bar{\sigma}'}{\tau} \left( \frac{1}{1 + \frac{1}{2}\eta(T_0 - t_0)} + \frac{1}{2} \frac{1}{(T - T_0)\eta} \right). \tag{64}$$

To have right hand side of (64) smaller then $\varepsilon_{G_{\mathcal{H}}}$ it is sufficient to choose $T_0$ and $T$ such that

$$\frac{4L^2\sigma\bar{\sigma}'}{\tau} \left( \frac{1}{1 + \frac{1}{2}\eta(T_0 - t_0)} \right) \leq \frac{1}{2}\varepsilon_{G_{\mathcal{H}}} \tag{65}$$

$$\frac{4L^2\sigma\bar{\sigma}'}{\tau}\left(\frac{1}{2}\frac{1}{(T-T_0)\eta}\right) \leq \frac{1}{2}\varepsilon_{G_{\mathcal{H}}} \tag{66}$$

Hence if $T_0 \geq t_0 + \frac{2}{\eta}\left(\frac{8L^2\sigma\bar{\sigma}'}{\tau\varepsilon_{G_{\mathcal{H}}}} - 1\right)$ and $T \geq T_0 + \frac{4L^2\sigma\bar{\sigma}'}{\tau\varepsilon_{G_{\mathcal{H}}}\eta}$ then (65) and (66) are satisfied.

$\square$

**Proposition 1** (Local Certificates). *Assume $g_i$ has L-bounded support, and let $\mathcal{N}_k := \{j : \mathcal{W}_{jk} > 0\}$ be the set of nodes accessible to node $k$. Then for any given $\varepsilon > 0$, we have*

$$G_{\mathcal{H}}(\mathbf{x}; \{\mathbf{v}_k\}_{k=1}^K) \leq \varepsilon,$$

*if for all $k = 1, \ldots, K$ the following two local conditions are satisfied:*

$$\mathbf{v}_k^\top \nabla f(\mathbf{v}_k) + \sum_{i \in \mathcal{P}_k} \left(g_i(\mathbf{x}_i) + g_i^*(-\mathbf{A}_i^\top \nabla f(\mathbf{v}_k))\right) \leq \frac{\varepsilon}{2K} \tag{9}$$

$$\left\|\nabla f(\mathbf{v}_k) - \frac{1}{|\mathcal{N}_k|}\sum_{j \in \mathcal{N}_k}\nabla f(\mathbf{v}_j)\right\|_2 \leq \left(\sum_{k=1}^K n_k^2 \sigma_k\right)^{-1/2}\frac{1-\beta}{2L\sqrt{K}}\varepsilon, \tag{10}$$

*Proof.* If the $\mathbf{w}_k$ variable in the duality gap (6) is fixed to $\mathbf{w}_k = \mathbf{g}_k := \nabla f(\mathbf{v}_k)$, then using the equality condition of the Fenchel-Young inequality on $f$, the duality gap can be written as follows

$$G_{\mathcal{H}}(\mathbf{x}; \{\mathbf{v}_k\}_{k=1}^K) := \sum_{k=1}^K\left(\langle\mathbf{v}_k, \mathbf{g}_k\rangle + \sum_{i \in \mathcal{P}_k}g_i(\mathbf{x}_i) + g_i^*(-\mathbf{A}_i^\top\bar{\mathbf{g}})\right) \tag{67}$$

where $\bar{\mathbf{g}} = \frac{1}{K}\sum_{k=1}^K\mathbf{g}_k$ is the only term locally unavailable.

$$G_{\mathcal{H}} \leq \sum_{k=1}^K\left(\langle\mathbf{v}_k, \mathbf{g}_k\rangle + \sum_{i \in \mathcal{P}_k}g_i(\mathbf{x}_i) + g_i^*(-\mathbf{A}_i^\top\mathbf{g}_k)\right) + \left|\sum_{k=1}^K\sum_{i \in \mathcal{P}_k}\left(g_i^*(-\mathbf{A}_i^\top\bar{\mathbf{g}}) - g_i^*(-\mathbf{A}_i^\top\mathbf{g}_k)\right)\right| \tag{68}$$

If both terms in (68) are less than $\varepsilon/2$, then $G_{\mathcal{H}} \leq \varepsilon$. Since the first term can be calculated locally, we only need for all $k = 1, \ldots K$

$$\langle\mathbf{v}_k, \mathbf{g}_k\rangle + \sum_{i \in \mathcal{P}_k}g_i(\mathbf{x}_i) + g_i^*(-\mathbf{A}_i^\top\mathbf{g}_k) \leq \frac{\varepsilon}{2K}. \tag{69}$$

Consider the second term in (68). Compute the difference between $g_i^*(-\mathbf{A}_i^\top\bar{\mathbf{g}})$ and $g_i^*(-\mathbf{A}_i^\top\mathbf{g}_k)$

$$|g_i^*(-\mathbf{A}_i^\top\bar{\mathbf{g}}) - g_i^*(-\mathbf{A}_i^\top\mathbf{g}_k)| \leq L|-\mathbf{A}_i^\top(\bar{\mathbf{g}} - \mathbf{g}_k)| \leq L\|\mathbf{A}_i\|_2\|\bar{\mathbf{g}} - \mathbf{g}_k\|_2 \tag{70}$$

where we use Lemma 3 and $L$-Lipschitz continuity. Then sum up coordinates $i \in \mathcal{P}_k$ on node $k$

$$\left|\sum_{i \in \mathcal{P}_k}\left(g_i^*(-\mathbf{A}_i^\top\bar{\mathbf{g}}) - g_i^*(-\mathbf{A}_i^\top\mathbf{g}_k)\right)\right| \leq L\|\bar{\mathbf{g}} - \mathbf{g}_k\|_2\sum_{i \in \mathcal{P}_k}\|\mathbf{A}_i\|_2. \tag{71}$$

Sum up (71) for all $k = 1, \ldots, K$ and apply the Cauchy-Schwarz inequality

$$\left|\sum_{k=1}^K\sum_{i \in \mathcal{P}_k}\left(g_i^*(-\mathbf{A}_i^\top\bar{\mathbf{g}}) - g_i^*(-\mathbf{A}_i^\top\mathbf{g}_k)\right)\right| \leq L\sqrt{\sum_{k=1}^K\|\bar{\mathbf{g}} - \mathbf{g}_k\|_2^2}\sqrt{\sum_{k=1}^K(\sum_{i \in \mathcal{P}_k}\|\mathbf{A}_i\|_2)^2}. \tag{72}$$

We will upper bound $\sum_{i \in \mathcal{P}_k}\|\mathbf{A}_i\|_2$ and $\|\bar{\mathbf{g}} - \mathbf{g}_k\|_2$ separately. First we have

$$\sum_{i \in \mathcal{P}_k}\|\mathbf{A}_i\|_2 \leq \sqrt{n_k}\|\mathbf{A}_{[k]}\|_F \leq n_k\|\mathbf{A}_{[k]}\|_{\infty,2} \leq n_k\sqrt{\sigma_k}, \tag{73}$$

where we write $\|.\|_{\infty,2}$ for the largest Euclidean norm of a column of the argument matrix, and then used the definition of $\sigma_k$ as in (7). Let us write $\mathbf{G} := [\mathbf{g}_1; \cdots; \mathbf{g}_K]$, $\mathbf{E} := \frac{1}{K}[\mathbf{1}; \cdots; \mathbf{1}]$, then apply Young's inequality with $\delta$

$$\sum_{k=1}^K\|\mathbf{g}_k - \bar{\mathbf{g}}\|_2^2 = \|\mathbf{G} - \mathbf{G}\mathbf{E}\|_F^2$$

$$\leq(1+\frac{1}{\delta})\left\|\mathbf{G}-\mathbf{G}\mathcal{W}\right\|_F^2+(1+\delta)\left\|\mathbf{G}\mathcal{W}-\mathbf{G}\mathbf{E}\right\|_F^2$$

$$=(1+\frac{1}{\delta})\left\|\mathbf{G}-\mathbf{G}\mathcal{W}\right\|_F^2+(1+\delta)\left\|\mathbf{G}(\mathbf{I}-\mathbf{E})(\mathcal{W}-\mathbf{E})\right\|_F^2$$

$$\leq(1+\frac{1}{\delta})\left\|\mathbf{G}-\mathbf{G}\mathcal{W}\right\|_F^2+(1+\delta)\beta^2\left\|\mathbf{G}(\mathbf{I}-\mathbf{E})\right\|_F^2$$

$$=(1+\frac{1}{\delta})\sum_{k=1}^K\left\|\mathbf{g}_k-\frac{1}{|\mathcal{N}_k|}\sum_{j\in\mathcal{N}_k}\mathbf{g}_j\right\|_2^2+(1+\delta)\beta^2\sum_{k=1}^K\left\|\mathbf{g}_k-\bar{\mathbf{g}}\right\|_2^2$$

Take $\delta:=(1-\beta)/\beta$, then we have

$$\sum_{k=1}^K\|\mathbf{g}_k-\bar{\mathbf{g}}\|_2^2\leq\frac{1}{1-\beta}\sum_{k=1}^K\left\|\mathbf{g}_k-\frac{1}{|\mathcal{N}_k|}\sum_{j\in\mathcal{N}_k}\mathbf{g}_j\right\|_2^2+\beta\sum_{k=1}^K\|\mathbf{g}_k-\bar{\mathbf{g}}\|_2^2$$

$$\leq\frac{1}{(1-\beta)^2}\sum_{k=1}^K\left\|\mathbf{g}_k-\frac{1}{|\mathcal{N}_k|}\sum_{j\in\mathcal{N}_k}\mathbf{g}_j\right\|_2^2 \tag{74}$$

We now use (73) and (74) and impose

$$\frac{1}{(1-\beta)^2}\sum_{k=1}^K\left\|\mathbf{g}_k-\frac{1}{|\mathcal{N}_k|}\sum_{j\in\mathcal{N}_k}\mathbf{g}_j\right\|_2^2\sum_{k=1}^K n_k^2\sigma_k\leq\left(\frac{\varepsilon}{2L}\right)^2 \tag{75}$$

then (72) is less than $\varepsilon/2$. Finally, (75) can be guaranteed by imposing the following restrictions for all $k=1,\dots,K$

$$\left\|\mathbf{g}_k-\frac{1}{|\mathcal{N}_k|}\sum_{j\in\mathcal{N}_k}\mathbf{g}_j\right\|_2^2\leq(1-\beta)^2\frac{\varepsilon^2}{4KL^2}\left(\sum_{k=1}^K n_k^2\sigma_k\right)^{-1} \tag{76}$$

$\square$

# D  Experiment details

In this section we provide greater details about the experimental setup and implementations. All the codes are written in PyTorch (0.4.0a0+cc9d3b2) with MPI backend [Paszke et al., 2017]. In each experiment, we run centralized CoCoA for a sufficiently long time until progress stalled; then use their minimal value as the approximate optima.

**DIGing.**  DIGing is a distributed algorithm based on inexact gradient and a gradient tracking technique. [Nedic et al., 2017] proves linear convergence of DIGing when the distributed optimization objective is strongly convex over time-varying graphs with a fixed learning rate. In this experiments, we only consider the time-invariant graph. The stepsize is chosen via a grid search. [Nedic et al., 2017] mentioned that the EXTRA algorithm [Shi et al., 2015] is almost identical to that of the DIGing algorithm when the same stepsize is chosen for both algorithms, so we only present with DIGing here.

**CoLa.**  We implement CoLa framework with local solvers from Scikit-Learn [Pedregosa et al., 2011]. Their ElasticNet solver uses coordinate descent internally. We note that since the framework and theory allow any internal solver to be used, CoLa could benefit even beyond the results shown by using existing fast solvers. We implement CoCoA as a special case of CoLa. The aggregation parameter $\gamma$ is fixed to 1 for all experiments.

**ADMM.**  Alternating Direction Method of Multipliers (ADMM) [Boyd et al., 2011] is a classical approach in distributed optimization problems. Applying ADMM to decentralized settings [Shi et al., 2014] involves solving

$$\min_{x_i,z_{ij}}\sum_{i=1}^L f_i(x_i)\qquad\text{s.t. }x_i=z_{ij},x_j=z_{ij},\qquad\forall\,(i,j)\in\mathcal{E}$$

Figure 5: The consensus violation curve of COLA in Figure 2.

Figure 6: Same settings as Figure 4 except that $\mathbf{x}_{[k]}$ are reset after node $k$ leaving the network.

where $z_{ij}$ is an auxiliary variable imposing the consensus constraint on neighbors $i$ and $j$. We therefore employ the coordinate descent algorithm to solve the local problem. The number of coordinates chosen in each round is the same as that of COLA. We choose the penalty parameter from the conclusion of [Shi et al., 2014].

**Additional experiments.** We provide additional experimental results here. First the *consensus violation* $\sum_{k=1}^{K} \|\mathbf{v}_k - \mathbf{v}_c\|_2^2$ curve for Figure 2 is displayed in Figure 5. As we can see, the consensus violation starts with 0 and soon becomes very large, then gradually drops down. This is because we are minimizing the sum of $\mathcal{H}_A^{(t)}$ and $\delta^{(t)}$, see the proof of Theorem 1. Then another model under failing nodes is tested in Figure 6 where $\mathbf{x}_{[k]}$ are initialized to 0 when node $k$ leave the network. Note that we assume the leaving node $k$ will inform its neighborhood and modify their own local estimates so that the rest nodes still satisfy $\frac{1}{\#\text{nodes}} \sum_k \mathbf{v}_k = \mathbf{v}_c$. This failure model, however, oscillates and does not converge fast.

# E    Details regarding extensions

## E.1    Fault tolerance and time varying graphs

In this section we extend framework COLA to handle fault tolerance and time varying graphs. Here we assume when a node leave the network, their local variables $\mathbf{x}$ are frozen. We use same assumptions about the fault tolerance model in [Smith et al., 2017].

**Definition 5** (Per-Node-Per-Iteration-Approximation Parameter)**.** *At each iteration $t$, we define the accuracy level of the solution calculated by node $k$ to its subproblem as*

$$\theta_k^t := \frac{\mathscr{G}_k^{\sigma'}(\Delta \mathbf{x}_k^{(t)}; \mathbf{x}_{[k]}^{(t)}, \mathbf{v}_k^{(t)}) - \mathscr{G}_k^{\sigma'}(\Delta \mathbf{x}_k^\star; \mathbf{x}_{[k]}^{(t)}, \mathbf{v}_k^{(t)})}{\mathscr{G}_k^{\sigma'}(\mathbf{0}; \mathbf{x}_{[k]}^{(t)}, \mathbf{v}_k^{(t)}) - \mathscr{G}_k^{\sigma'}(\Delta \mathbf{x}_k^\star; \mathbf{x}_{[k]}^{(t)}, \mathbf{v}_k^{(t)})} \tag{77}$$

*where $\Delta \mathbf{x}_k^\star$ is the minimizer of the subproblem $\mathscr{G}_k^{\sigma'}(\cdot; \mathbf{x}_{[k]}^{(t)}, \mathbf{v}_k^{(t)})$. We allow this value to vary between [0, 1] with $\theta_k^t := 1$ meaning that no updates to subproblem $\mathscr{G}_k^{\sigma'}$ are made by node $k$ at iteration $t$.*

The flexible choice of $\theta_k^t$ allows the consideration of stragglers and fault tolerance. We also need the following assumption on $\theta_k^t$.

**Assumption 2** (Fault Tolerance Model)**.** *Let $\{\mathbf{x}^{(t)}\}_{t=0}^T$ be the history of iterates until the beginning of iteration $T$. For all nodes $k$ and all iterations $t$, we assume $p_k^t := \mathcal{P}[\theta_k^t = 1] \le p_{max} < 1$ and $\hat{\Theta}_k^T := \mathbb{E}[\theta_k^T | \{\mathbf{x}^{(t)}\}_{t=0}^T, \theta_k^T < 1] \le \Theta_{max} < 1$.*

In addition we write $\bar{\Theta} := p_{\max} + (1 - p_{\max})\Theta_{max} < 1$. Another assumption on time varying model is necessary in order to maintain the same linear and sublinear convergence rate. It is from [Nedic et al., 2017, Assumption 1]:

**Assumption 3** (Time Varying Model). *Assume the mixing matrix $\mathcal{W}(t)$ is a function of time $t$. There exist a positive integer $B$ such that the spectral gap satisfies the following condition*

$$\sigma_{max}\left\{\prod_{i=t}^{t+B-1}\mathcal{W}(i) - \tfrac{1}{K}\mathbf{1}\mathbf{1}^\top\right\} \le \beta_{max} \qquad \forall\, t \ge 0.$$

We change the Algorithm 1 such that it performs gossip step for $B$ times between solving subproblems. In this way, the convergence rate on time varying mixing matrix is similar to a static graph with mixing matrix $\prod_{i=t}^{t+B-1}\mathcal{W}(i)$. The sublinear/linear rate can be proved similarly.

## E.2 Data dependent aggregation parameter

**Definition 6** (Data-dependent aggregation parameter). *In Algorithm 1, the aggregation parameter $\gamma$ controls the level of adding $\gamma$ versus averaging $\gamma := \frac{1}{K}$ of the partial solution from all machines. For the convergence discussed below to hold, the subproblem parameter $\sigma'$ must be chosen not smaller than*

$$\sigma' \ge \sigma'_{min} := \gamma \max_{\mathbf{x}\in\mathbb{R}^n} \frac{\|\mathbf{A}\mathbf{x}\|^2}{\sum_{k=1}^{K}\left\|\mathbf{A}\mathbf{x}_{[k]}\right\|^2} \tag{78}$$

*The simple choice of $\sigma' := \gamma K$ is valid for (78), closer to the actual bound given in $\sigma'_{min}$.*

## E.3 Hessian subproblem

If the Hessian matrix of $f$ is available, it can be used to define better local subproblems, as done in the classical distributed setting by [Gargiani, 2017, Lee and Chang, 2017, Dünner et al., 2018, Lee et al., 2018]. We use same idea in the decentralized setting, defining the improved subproblem

$$\begin{aligned}
\mathscr{G}_k^{\sigma'}(\Delta\mathbf{x};\mathbf{x}_{[k]},\mathbf{v}_k) :=& \tfrac{1}{K}f(\mathbf{v}_k) + \left\langle \textstyle\sum_{l=1}^{K}\mathcal{W}_{kl}\nabla f(\mathbf{v}_l), \mathbf{A}\Delta\mathbf{x}_{[k]}\right\rangle \\
&+ \tfrac{1}{2}(\mathbf{A}\Delta\mathbf{x}_{[k]})^\top \left(\textstyle\sum_{l=1}^{K}\mathcal{W}_{kl}\nabla^2 f(\mathbf{v}_l)\right)\mathbf{A}\Delta\mathbf{x}_{[k]} + \sum_{i\in\mathcal{P}_k} g_i(x_i + \Delta x_i)
\end{aligned} \tag{79}$$

The sum of previous subproblems satisfies the following relations

$$\begin{aligned}
&\sum_{k=1}^{K}\mathscr{G}_k^{\sigma'}(\mathbf{0};\mathbf{x}_{[k]}^{(t+1)},\mathbf{v}_k^{(t+1)})\\
=&\frac{1}{K}\sum_{k=1}^{K} f\left(\sum_{l=1}^{K}\mathcal{W}_{kl}\mathbf{v}_l^{(t)} + \gamma K\mathbf{A}\Delta\mathbf{x}_{[k]}\right) + \sum_{i\in\mathcal{P}_k} g_i(x_i^{(t)} + (\Delta\mathbf{x}_{[k]})_i)\\
\le&\frac{1}{K}\sum_{k=1}^{K}\sum_{l=1}^{K}\mathcal{W}_{kl}\left\{f(\mathbf{v}_l^{(t)}) + \langle\nabla f(\mathbf{v}_l^{(t)}),\gamma K\mathbf{A}\Delta\mathbf{x}_{[k]}\rangle + \frac{1}{2}(\mathbf{A}\Delta\mathbf{x}_{[k]})^\top\nabla^2 f(\mathbf{v}_l^{(t)})\mathbf{A}\Delta\mathbf{x}_{[k]}\right\}\\
&+ \sum_{i\in\mathcal{P}_k} g_i(x_i^{(t)} + (\Delta\mathbf{x}_{[k]})_i)\\
=&\sum_{k=1}^{K}\mathscr{G}_k^{\sigma'}(\Delta\mathbf{x};\mathbf{x}_{[k]}^{(t)},\mathbf{v}_k^{(t)}) \le \sum_{k=1}^{K}\mathscr{G}_k^{\sigma'}(\mathbf{0};\mathbf{x}_{[k]}^{(t)},\mathbf{v}_k^{(t)})
\end{aligned}$$

This means that the sequence $\left\{\sum_{k=1}^{K}\mathscr{G}_k^{\sigma'}(\mathbf{0};\mathbf{x}_{[k]}^{(t)},\mathbf{v}_k^{(t)})\right\}_{t=0}^{\infty}$ is monotonically non-increasing. Following the reasoning in this paper, we can have similar convergence guarantees for both strongly convex and general convex problems. Formalizing all detailed implications here would be out of the scope of this paper, but the main point is that the second-order techniques developed for the COCOA framework also have their analogon in the decentralized setting.