[Reviews · NeurIPS 2018]

Reviewer 1



This paper deals with learning linear models in a decentralized setting, where each node holds a subset of the dataset (features or data points, depending on the application) and communication can only occur between neighboring nodes in a connected network graph. The authors extend the CoCoA algorithm, originally designed for the distributed (master/slave) setting. They provide convergence rates as well as numerical comparisons. The authors should state more clearly that they are extending CoCoA to the decentralized setting. The adaptation of the setup, the local subproblems and the algorithm itself are fairly direct by restricting the information accessible by each node to its direct neighbors (instead of having access to information from all nodes). Despite the limited originality in the algorithm design, the main technical contribution is the derivation of convergence rates for the decentralized setting. This non-trivial result makes the paper an interesting contribution to the literature of decentralized optimization. In particular, the resulting algorithm has several properties (inherited from CoCoA) that are useful in the decentralized setting. I have a few technical questions: 1/ Can you clarify and comment the dependence of the network topology (spectral gap) on the convergence rates of Thm 1-3? A typical dependence is on the inverse of the square root of the spectral gap (see e.g. [1, 2]). In Thm 2, why is there no dependence on the spectral gap? 2/ CoCoA achieves linear convergence even when only g_i is strongly convex (not f), but this decentralized version only achieves sublinear convergence in this case (Theorem 2). Is there a fundamental reason for this or is it due to the proof technique? 3/ The convergence guarantees hold for the quantity \bar{x}, which is an average over several iterations. As each node holds a block of the iterate x^(t) at each iteration, it looks like making the model \bar{x} available to all nodes as learning progresses (so they can use the best current model to make predictions) could require a lot of additional communication. Is there a way around this? Regarding the experiments, experimenting with only 16 nodes is quite disappointing for the decentralized setting, which is largely motivated by the scalability to large networks. It would be much more convincing if the authors can show that their algorithm still behaves well compared to competitors on networks with many more nodes (this can be done by simulation if needed). The clarity of the presentation can be improved: - The authors remain very elusive on what the matrix A represents in practice. They do not explicitly mention that depending on the task, the dataset must be split sample-wise or feature-wise (sometimes, both are possible if both the the primal and the dual match the assumptions). Giving a few concrete examples would really help the reader understand better the possible application scenarios. - This aspect is also very unclear in the experiments. For each task, what is distributed? And how are features/samples distributed across nodes? - It would be useful to give an interpretation of the form of the subproblem (3). Other comments/questions: - Lines 33-34: most decentralized optimization algorithms for sum-structured problems do not rely on an i.i.d. assumption or completely fail when it is violated (but they can of course be slower). - Line 139: the dependence of the communication cost on d (which can be the number of features or the total number of samples) should be made clear. Depending on the task and dataset, this dependence on d may make the algorithm quite inefficient in communication. - How was the rho parameter of ADMM set in the experiments? - It is false to argue that the proposed algorithm does not have any parameter to select. At the very least, one should carefully choose \Theta, the subproblem approximation parameter. There may be additional parameters for the local solver. Typos: - Before eq (2): "Let set" --> "Let the set" - Line 132: "Appendix E.2" --> "Appendix E.1" - Line 178: missing tilde on G - Line 180: we recovers References: [1] Duchi et al. Dual Averaging for Distributed Optimization: Convergence Analysis and Network Scaling. IEEE TAC 2012. [2] Colin et al. Gossip Dual Averaging for Decentralized Optimization of Pairwise Functions. ICML 2016. ============ After Rebuttal ============ Thanks for the clarifications and updated results.

Reviewer 2



The authors consider the problem learning a (regularized) linear model over a graph of agents. At each iteration, the agents exchange with their neighbors and perform an (approximate) proximal gradient step. They provide linear and sublinear rates depending on the strong convexity of the functions. Despite interesting ideas, including the rate derivations, this papers lack connections with the literature and some clarifications in its present form. [Distributed Setup] The distributed setup is rather well explained although it is not exactly standard as the data matrix is split by the *columns* (i.e. the features) among the agents; usually, the lines/examples are split. This difference should be more clearly mentioned and motivated in a practical context (near l64 page 2). As the simulations are performed on a single machine, one would expect a justification for considering this kind of data splitting and network topology. * Then, the communications are done using a doubly-stochastic mixing matrix as commonly accepted in Gossip theory. The paragraph before Assumption 1 is unclear: it is well known that as soon as an (undirected) graph is connected, the spectral gap is positive, this would make assumption 1 more readable. [Optimization algorithm] As the authors mention (Sec 1.2, e.g. 108), the proposed algorithm falls into the primal-dual methods. * At first glance, it seems close (as the matrix A cannot be inverted) to Condat-Vu's algorithm (Condat, L. (2013). A primal–dual splitting method for convex optimization involving Lipschitzian, proximable and linear composite terms. Journal of Optimization Theory and Applications, 158(2), 460-479.) Decentralized algorithms based on this algorithm were proposed, see e.g. Bianchi, P., Hachem, W., & Iutzeler, F. (2016). A coordinate descent primal-dual algorithm and application to distributed asynchronous optimization. IEEE Transactions on Automatic Control, 61(10), 2947-2957. The synchronous version (DADMM+ in their paper) looks indeed close to what you propose and should definitively be compared to the proposed method. * Line 5 in the algorithm you approximatively solve problem (2), which leads to two concerns: (i) why: minimizing (2) seems equivalent to a proximity operation on the corresponding g_i on a point that is a gradient step on the coordinates of f with stepsize tau/sigma'; this seems to be computable exactly quite cheaply especially for the Lasso and Ridge problems considered in the experiments; (ii) how: I don't get how to obtain approximate solutions of this problem apart from not computing some coordinates but then theta can be arbitrarily bad and so are the rates. * I do not see a discussion on how to choose gamma. Furthermore, it seems rather explicit that the gradient descent is performed with the inverse of the associated Lipschitz constant. [Convergence rates] The derivations seem valid but their significance is kind of obfuscated. * In Theorem 1, the assumption that g has a bounded support (or rather that the iterates stay in a bounded subspace) is rather strong but ok, what bothers me is that (i) it is hard to decipher the rate, a simple case with theta = 0, gamma =1 might help. In addition, the assumption on rho also would require some explanations. * In Theorem 2, it look that we actually have the usual conditioning/T rate but clarifications are also welcome * In Theorem 3, two conditions bother me: (i) mu tau >= D_1 seem like a minimal conditioning of the problem to have a linear rate; (ii) the condition of beta seems to look like a condition on how well connected the graph has to be to get the linear rate. Again, this is not discussed. Minor comments: * l91 "bridgin"G * l232: the authors say "CoLa is free of tuning parameters" but the gradient descent is actually natively performed with the inverse Lipschitz constant which is a possibility in most algorithms of the literature (and can be local as extended in Apx E1). Furthermore, there is also the parameter gamma to tune, theta to adapt... ============ After Rebuttal ============ After reading the author feedback and the other reviews, I upgraded my score to 6. The main reasons for that upgrade are: * the improvement of their main theorem to any connected graph * a misconception on my side for the difficulty of pb. (2) However, it still seems important to me to clarify the non-triviality of (2), mention the relations between CoCoA and ADMM, and clarify that gamma and Theta are not hyper-parameter (cf l. 12-14 of the feedback).

Reviewer 3



This paper essentially proposed a decentralized version of CoCoA algorithm, or a decentralized version of block coordinate descent. While I enjoyed reading the first section, the theory section is a little bit over complicated in term of presentation. Authors may consider how to simplify notations and statements. Despite of this, I have a few questions / concerns that need authors to respond in rebuttal - A few key theoretical results are not clear enough to me. In particular, how does the proposed decentralized algorithm improves CoCoA in theory? Authors need to compare the communication cost and the computation complexity at least. (Authors show that the convergence rate is sort of consistent with CoCoA in the extreme case, but that may not be enough. People may want to see if your method has any advantage over CoCaA in theory by jointly considering communication and computation cost.) - The speedup property is not clear to. Since the notation and the statement of the theoretical results is over complicate, it is hard to see if more workers will accelerate the training process and how? - The experiments are performed on the multi-core machine, where the communication is not the bottleneck. To my personal experiences, the decentralized algorithm may not have advantage over centralized algorithms. ============= Authors partially addressed my concerns. The comparison to centralized methods are also expected to include since in many ML application, people can choose either centralized network or decentralized network. Authors may refer to the comparison in [14].